# Adversarial Robustness with Non-uniform Perturbations

**Ecenaz Erdemir**
Imperial College London
e.erdemir17@imperial.ac.uk

**Jeffrey Bickford**
Amazon Web Services
jbick@amazon.com

**Luca Melis**
Amazon Web Services
lucmeli@amazon.com

**Sergül Aydöre**
Amazon Web Services
saydore@amazon.com

## Abstract

Robustness of machine learning models is critical for security related applications, where real-world adversaries are uniquely focused on evading neural network based detectors. Prior work mainly focus on crafting adversarial examples (AEs) with small uniform norm-bounded perturbations across features to maintain the requirement of imperceptibility. However, uniform perturbations do not result in realistic AEs in domains such as malware, finance, and social networks. For these types of applications, features typically have some semantically meaningful dependencies. The key idea of our proposed approach[1] is to enable non-uniform perturbations that can adequately represent these feature dependencies during adversarial training. We propose using characteristics of the empirical data distribution, both on correlations between the features and the importance of the features themselves. Using experimental datasets for malware classification, credit risk prediction, and spam detection, we show that our approach is more robust to real-world attacks. Finally, we present robustness certification utilizing non-uniform perturbation bounds, and show that non-uniform bounds achieve better certification.

## 1 Introduction

Deep neural networks (DNNs) are commonly used in a wide-variety of security-critical applications such as self-driving cars, spam detection, malware detection and medical diagnosis [1]. However, DNNs have been shown to be vulnerable to adversarial examples (AEs), which are perturbed inputs designed to fool the machine learning systems [2–4]. To mitigate this problem, a line of research has focused on adversarial robustness of DNNs as well as the certification of these methods [1, 5–10].

Adversarial training (AT) is one of the most effective empirical defenses against adversarial attacks [1, 11]. The goal during training is to minimize the loss of the DNN when perturbed samples are used. This way, the model becomes robust to real-world adversarial attacks. Though these empirical defenses do not provide theoretically provable guarantees, they have been shown to be robust against the strongest known attacks [12]. Some of the most common state-of-the-art adversarial attacks, such as projected gradient descent (PGD) [1] and fast gradient sign method (FGSM) [12], perturb training samples under a norm-ball constraint to maximize the loss of the network. The goal of certification, on the other hand, is to report whether an AE exists within an $\ell_p$ norm centered at a given sample with a fixed radius. Certified defense approaches introduce theoretical robustness guarantees against norm-bounded perturbations [8, 9, 13, 14].

---

[1]Code is available at https://github.com/amazon-research/adversarial-robustness-with-nonuniform-perturbations

35th Conference on Neural Information Processing Systems (NeurIPS 2021).

In the computer vision domain, the adversary's goal is to generate perturbed images that cause misclassifications in a DNN. It is often assumed that limiting a uniform norm-ball constraint results in perturbations that are imperceptible to the human eye. However in other applications such as fraud detection [15], spam detection [16], credit card default prediction [17, 18] and malware detection [19–21], norm-bounded uniform perturbations may result in unrealistic transformations. Perturbed samples must comply with certain constraints related to the domain, hence preventing us from borrowing these assumptions from computer vision. These constraints can be on semantically meaningful feature dependencies, expert knowledge of possible attacks, and immutable features [20, 22]. This paper proposes a methodology to generate non-uniform perturbations that takes into account the characteristics of the empirical data distribution. Our results demonstrate that these non-uniform perturbations outperform uniform norm-ball constraints in these types of applications.

## 1.1 Background and Motivation

AT is a min-max problem minimizing the DNN loss which is maximized by adversarial perturbations (call $\delta$). State-of-the-art approaches for optimizing $\delta$ usually assume that all the input features require equal levels of robustness, however, this might not be the case for many applications. A toy example for AT with non-uniform perturbations is given in Appendix A.1. The intuition behind the need for non-uniform constraints is apparent across many industrial applications. A common cybersecurity application is malware detection, which identifies if an executable file is benign or malicious. Unlike images, diverse and semantically meaningful features are extracted from the executable file and are passed to a machine learning model. To maintain the functionality of an executable file during an adversarial attack, certain features may be immutable and perturbations may result in an unrealistic scenario. For example in the Android malware space, application permissions, such as permission to access a phone's location service, are required for malicious functionality and cannot be perturbed [19]. In a finance scenario where customer credit card applications are evaluated by machine learning models, a possible set of features include age, gender, income, savings, education level, number of dependents, etc. In this type of dataset there are clear dependencies between features, for example the number of dependents has a meaningful correlation with age. When detecting spammers within social networks, features are extracted from accounts and may include the length of the username, length of user description, number of following and followers as well as the ratio between them, percentage of bidirectional friends, etc. Similar to the previous finance example, there is a meaningful correlation between features such as the percentage of bidirectional friends and the ratio of followers.

In all of these scenarios, non-uniform perturbations can be used to maintain these correlations and semantically meaningful dependencies resulting in more realistic AEs. In this work, we propose adversarial training with these more realistic perturbations to increase the robustness against real-world adversarial attacks. Specifically, our contributions are: (i) Instead of considering an allowed perturbation region where all the features are treated uniformly, i.e., $\|\delta\|_p \leq \epsilon$, we consider a transformed input perturbation constraint, i.e., $\|\Omega\delta\|_p \leq \epsilon$ where $\Omega$ is a transformation matrix, which takes the available information into account, such as feature importance, feature correlations and/or domain knowledge. Hence, the transformation in the norm ball constraint results in non-uniform input perturbations over the features. (ii) For various applications such as malware detection, credit risk prediction and spam detection, we show that robustness using non-uniform perturbations outperforms the commonly-used uniform approach. (iii) To provide provable guarantees for non-uniform robustness, we modify two known certification methods, linear programming and randomized smoothing, to account for non-uniform perturbation constraints.

## 1.2 Related Work

Different levels of robustness of different features have already been studied in the literature [23–25]. Among these, [23] and [24] study the effect of robust and non-robust features in standard and adversarial training. Both works show that different types of features might either be vulnerable or robust to small input perturbations. This is a discrete interpretation of different levels of perturbation tolerance which is the core idea of our approach. Both works provide effective theoretical analysis of robust and non-robust features and their effect on clean and adversarial accuracy, while we propose a defense mechanism utilizing this phenomenon.

A closely related work to ours is [25] which considers non-uniform perturbation bounds for input features. Given a non-uniform adversarial budget $\epsilon$ for $\ell_\infty$ norm bounded inputs, [25] proposes

a framework that maximizes the volumes of certified bounds. However, this work only proposes robustness certification of pre-trained models, and does not consider training a robust model against these non-uniform perturbations. The approach is also data-agnostic meaning that it does not take correlations or an additional knowledge on the data into account. We instead use non-uniform perturbations in data-dependent AT to achieve robustness in a DNN model. In fact, [25] mentioned consideration of feature correlations as a potential future direction of their work.

In [26], a conditional variational autoencoder (CVAE) is trained to learn perturbation sets for image data and the generated adversarial images are then used in augmented training. The work proposes robustness against common image corruptions as well as $\ell_p$ perturbations. However, it mainly focuses on possible corruptions and attacks specific to image domain, and assumes access to the test data distribution. In our work, we focus on non-image data which have correlated and different scale features and we do not rely on an additional knowledge of the test set.

Recent work in the malware detection space [20] creates realistic AEs by defining a set of comprehensive and realistic constraints on how an input file can be transformed. Though this approach creates realistic AEs, collecting a representative corpus of raw input files is a challenging problem [27]. The recent release of binary feature sets, such as the EMBER dataset [28], enables model development without having access to the raw input files. Our work can be used to create more realistic AEs in situations where access to a large representative corpus of files is not possible.

## 2 Non-uniform Adversarial Perturbations

In adversarial training, the worst case loss for an allowed perturbation region is minimized over parameters of a function representing a DNN. The objective of the adversary can be written as the inner maximization of adversarial training:

$$\underset{\delta \in \Delta_{\epsilon,p}}{\text{maximize}} \quad \ell(f_\theta(x + \delta), y), \tag{1}$$

where $x \in \mathcal{X}$ and $y \in \mathcal{Y}$ are dataset inputs and labels, $\Delta_{\epsilon,p} = \{\delta : \|\delta\|_p \leq \epsilon\}$ is an $\ell_p$ ball of radius $\epsilon$ which defines the feasible perturbation region. Standard PGD follows steepest descent which iteratively updates $\delta$ in the gradient direction to increase the loss:

$$\delta^{t+1} = \delta^t + \alpha \frac{\nabla_\delta \ell(f_\theta(x + \delta^t), y)}{\|\nabla_\delta \ell(f_\theta(x + \delta^t), y)\|_p} \tag{2}$$

at iteration $t$, and then it projects $\delta$ to the closest point onto the $\ell_p$ ball:

$$\mathcal{P}_{\Delta_{\epsilon,p}}(\delta) := \underset{\delta' \in \Delta_{\epsilon,p}}{\arg\min} \|\delta - \delta'\|_2^2 = \epsilon \frac{\delta}{max\{\epsilon, \|\delta\|_p\}} \tag{3}$$

where the distance between $\delta$ and $\delta'$ is the Euclidean distance, and the projection corresponds to normalizing $\delta$ to have a maximum $\ell_p$ norm which is equal to $\epsilon$. Now, we introduce an adversarial constraint set that non-uniformly limits adversarial variations in different, potentially correlated dimensions, by

$$\tilde{\Delta}_{\epsilon,p} = \{\delta : \|\Omega\delta\|_p \leq \epsilon\} \tag{4}$$

where $\Omega \in \mathbb{R}^{d \times d}$. In our approach, $\delta$ is updated by equation (2) similar to the standard PGD, however, it is projected back to a non-uniform norm ball satisfying $\|\Omega\delta\|_p \leq \epsilon$. The corresponding projection operator will then be:

$$\mathcal{P}_{\tilde{\Delta}_{\epsilon,p}}(\Omega\delta) = \begin{cases} \epsilon \frac{\delta}{\|\Omega\delta\|_p} & if \quad \|\Omega\delta\|_p > \epsilon \\ \delta & \text{otherwise.} \end{cases} \tag{5}$$

The choice of $\Omega$ depends on how we model the expert knowledge or feature relationships. The following are our choices for the non-uniform perturbation sets.

### 2.1 Mahalanobis Distance (MD)

Euclidean distance between two points in a multi-dimensional space is a useful metric when the vectors have isotropic distribution (i.e. radially symmetric). This is because the Euclidean distance assumes each dimension has same scale (or spread) and are uncorrelated to other dimensions.

However, isotropy is usually not the case for real datasets in which different features might have different scales and can be correlated. Fortunately, MD accounts for how the features are scaled and correlated to one another [29]. Hence, it is a more useful metric if the data has non-isotropic distribution.

By formal definition, MD between vectors $z, z' \in \mathbb{R}^d$ is denoted by $d_M(z, z'|M) := \sqrt{(z - z')^T M (z - z')}$, where $M \in \mathbb{R}^{d \times d}$ is a positive semi-definite matrix which can be decomposed as $M = U^T U$, for $U \in \mathbb{R}^{d \times d}$. The dissimilarity between two vectors from a distribution with covariance $\Sigma$ can be measured by selecting $M = \Sigma^{-1}$. If feature vectors of a dataset are uncorrelated and have unit variances, their covariance matrix is $\Sigma = I$, which reduces their MD to Euclidean distance.

We are interested in the distance between the original and the perturbed sample. Since we assume all perturbations are additive, as common practice, the distance term we consider is $\sqrt{\delta^T M \delta}$. For a generalized MD in $\ell_p$ norm, selecting $\Omega = U^T$ corresponds to the perturbation set $\tilde{\Delta}_{\epsilon, p} = \{\delta : \|U^T \delta\|_p \leq \epsilon\}$ which generates AEs with feature correlations similar to the original dataset.

Robustness of an adversarially trained model is directly related to how realistic the generated AEs are during training. Now, we explore implications of selecting $\ell_2$ MD to define the limits of the perturbation set. To ensure the validity of the AEs, we consider the notion of consistency of the generated sample with real samples. [18] introduced the notion of $\epsilon$-inconsistency to quantify how likely an AE is. With slight change in their notation, we define $\gamma$-consistency as follows:

**Definition 2.1.** *For a consistency threshold $\gamma > 0$, an AE is $\gamma$-consistent if $f(x \mid y) \geq \gamma$, where $x \in \mathbb{R}^d$, and $f$ is a probability density function of a conditional Gaussian distribution with zero mean and covariance matrix $\Sigma_y$.*

**Theorem 2.1.** *If the AEs are generated according to MD constraint, then their $\gamma$-consistency has a direct relation to $\epsilon$ such that*

$$0 < \sqrt{2C - 2 \log \gamma} \leq \epsilon. \tag{6}$$

*where $C = -\log(2\pi)^{d/2} |\Sigma_y|^{1/2}$, $d$ is the dimension of $x$, and $\sqrt{\delta^T \Sigma_y^{-1} \delta} \leq \epsilon$.*

Theorem 2.1 implies that there is a direct relationship between limiting the MD of $\delta$ and ensuring consistent samples when the data is Gaussian. In other words, when the $\ell_2$ MD of the perturbations gets smaller, AEs become more consistent. See Appendix A.4 for the proof.

## 2.2 Weighted Norm

When $\Omega$ is a diagonal matrix, inner maximization constraint simply becomes the weighted norm of $\delta$ limited by $\epsilon$, and the weights are denoted by $\{\Omega_{i,i}\}_{i=1}^d$. Projection of $\delta$ under the new constraint corresponds to projection onto an $\ell_p$ norm ball of radius $\frac{\epsilon}{\Omega_{i,i}}$ for $i^{th}$ feature. These weights can be chosen exploiting domain, attack or model knowledge. For instance, more important features can be allowed to be perturbed more than the other features which have less effect on the output score of the classifier. This knowledge might come from Pearson's correlation coefficients [17] between the features of the training data and the corresponding labels, or Shapley values [30] for each feature.

Using Pearson's correlation coefficient of each feature with the corresponding target variable, i.e., $|\rho_{i,y}|$ for $i^{th}$ feature and output $y$, we let larger perturbation radii for more correlated features with the output. Due to the inverse relation between $\Omega_{i,i}$ and the radius of the norm ball, for $\bar{\rho}_{i,y} = \frac{1}{|\rho_{i,y}|}$ we select $\Omega = \frac{diag(\{\bar{\rho}_{i,y}\}_{i=1}^d)}{\|\{\bar{\rho}_{i,y}\}_{i=1}^d\|_2}$. Similarly, using Shapley values to represent feature importance, we define $\bar{s}_i = \frac{1}{|s_i|}$, where $s_i$ is the Shapley value of feature $i$. Then, we choose $\Omega = \frac{diag(\{\bar{s}_i\}_{i=1}^d)}{\|\{\bar{s}_i\}_{i=1}^d\|_2}$ by following the intuition that more important features should have larger perturbation radii.

In the malware domain, expert knowledge might help to rule out specific type of attacks crafted on immutable features due to feasibility constraints. This can be modelled by the proposed weighted norm constraint as masking the perturbations on immutable features. Hence, non-uniform perturbation approach enables various transformations on the attack space for robustness against realistic attacks.

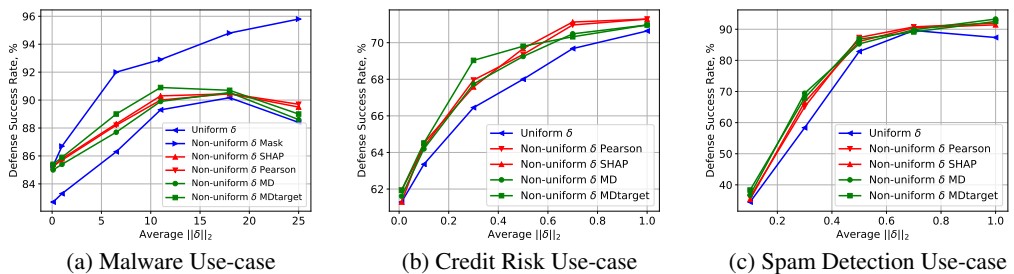

Figure 1: Defense success rate of $\ell_2$-PGD AT against the problem-space attacks, where all non-uniform perturbation defense approaches outperform the uniform approach for all use-cases.

## 3 Experimental Results

Here, we present experimental results to evaluate robustness of DNNs against adversarial attacks for binary classification problems on three applications: malware detection, credit risk prediction, and spam detection. We compare PGD with non-uniform perturbations during AT with PGD proposed in [1] based on uniform perturbations. For all applications, we evaluate our defense mechanisms on adversarial attacks proposed by other works. We use a machine with an Intel Xeon E5-2686 v4 @ 2.3 GHz CPU, and 4 Nvidia Tesla V100 GPUs. Details of the DNN architecture and pre-processing are given in A.3. Note that our goal is not to design the best possible neural network but instead compare the uniform perurbations [1] with various non-uniform perturbations during AT for a given DNN.

**Adversarial training (AT):** We perform AT in all use-cases by applying $\ell_2$-norm PGD for uniform perturbation sets, i.e., $\Delta_{\epsilon_1,2} = \{\delta : \|\delta\|_2 \leq \epsilon_1\}$, and non-uniform perturbation sets, i.e., $\tilde{\Delta}_{\epsilon_2,2} = \{\delta : \|\Omega\delta\|_2 \leq \epsilon_2\}$. Since potential adversaries are not interested in fooling the classifiers with negative class (target class) samples, $\delta$ perturbations are only applied to the positive classes during AT as commonly used especially in malware detection [20]. Positive classes are the malicious class in malware detection, bad class in credit risk prediction, and spammer class in spam detection. Moreover, for the sake of clean accuracy within the positive class, adversarial perturbations are applied to $90\%$ of the positive samples during training. Such hybrid approach where a weighted clean adversarial loss are optimized at once is common in literature [31].

To model the expert knowledge with diagonal $\Omega$, we use Pearson's correlation coefficient, Shapley values, and masking to allow perturbation only in mutable features. To compute Shapley values, we use SHAP [32] which utilizes a deep learning explainer. We also consider AT under the MD constraint, and select $\Omega = U^T$ such that $U^T U = \Sigma_y^{-1}$ considering two cases; $\Sigma_y$ is the covariance matrix of the entire training data, i.e., $y=\{0,1\}$, and $\Sigma_y$ is only for the negative (target) class $y=0$. We call the models after AT with non-uniform perturbations according to their $\Omega$ selection, e.g., *NU-$\delta$-Pearson* for Pearson's coefficients, *NU-$\delta$-SHAP* for Shapley values, *NU-$\delta$-Mask* for masking, *NU-$\delta$-MD* for MD using full covariance matrix and *NU-$\delta$-MDtarget* for MD using the covariance for only $y=0$. The choice $\Omega = I$ corresponds to AT with uniform perturbation constraint, which we call *Uniform-$\delta$*.

### 3.1 Malware Use-case

First, we consider a binary classification problem for malware detection using the EMBER dataset [28]. EMBER is a feature-based public dataset which is considered a benchmark for Windows malware detection. It contains 2381 features extracted from Windows PE files: $600K$ labeled training samples and $200K$ test samples. We refer to Appendix A.2 for more detailed description of the EMBER dataset features. Given a malware sample, an adversary's goal is to make the DNN conclude that a malicious sample is benign. We also consider PDF malware detection. We use the extracted features of the PDF malware classification dataset and its attacked samples provided in [33]. The repository contains 110841 samples with 135 features that are extracted by PDFrate-R [34].

**Attacks used for evaluation:** In the malware domain, test-time evasion attacks can be classified as *feature-space* and *problem-space* attacks. While the former crafts AEs by modifying the features extracted from binary files, the latter directly modifies malware binaries making sure of the validity

and inconspicuousness of the modified object. We evaluate the robustness of our model against evasion attacks which are crafted in problem-space, i.e., on PE files. For Windows malware, we incorporate the most successful attacks [35] from the machine learning static evasion competition [36]. Since the EMBER dataset only contains the extracted features of a file, a subset of malware binaries used for AE generation are obtained from VirusTotal [37] using the SHA-256 hash as identifier.

We observe that these problem-space attacks, which add various bytes to a file without modifying the core functionality, affect only the feature groups "Byte Histogram", "Byte Entropy Histogram" and "Section Information". Experts aware of these byte padding attacks understand which features can be manipulated by an attacker. In addition to the previous AT methods, we represent this *best case expert knowledge* by $\Omega = I_{mask}$, which is an identity matrix with non-zero diagonal elements only for "Byte Histogram", "Byte Entropy Histogram" and "Section Information" features. That is, the model is trained using PGD perturbations applied only to these features, and we call it *NU-$\delta$-Mask*. Our masking approach for the immutable features is similar to the *conserved features* in [38].

For PDF malware classification, we use a problem space attack called EvadeML [39]. It allows adding, removing and swapping objects, hence it is a stronger attack than most other problem space attacks in the literature, which typically only allow addition to preserve the malicious functionality.

**Numeric results:** To make a fair comparison between uniform and non-uniform approaches, $\epsilon$ for each method is selected such that their average distortion budgets, i.e., $\|\delta\|_2$, are approximately equal. For Windows malware classification, we test the detection success of adversarially trained models with 9000 AE sets generated by the problem-space attacks described in Appendix A.6. Figure 1a illustrates the average defense success rate against various problem-space attacks and shows that non-uniform perturbation approaches outperform the uniform perturbation in all cases. Moreover, *NU-$\delta$-MDtarget* performs closest to the best case expert knowledge *NU-$\delta$-Mask* for all cases except when $\|\delta\|_2 = 25$. The advantage of selecting $\Sigma$ from benign samples versus selecting from the entire dataset is that the direction of perturbations are led towards the target class, i.e. benign samples, for *NU-$\delta$-MDtarget*. We also do not observe a significant performance difference between *NU-$\delta$-Pearson* and *NU-$\delta$-SHAP*, while *NU-$\delta$-MD* only differs from the two for $\|\delta\|_2 = 25$. We refer to Table A.1 for detailed attack performances and to Table A.2 for defense S.R. results with clean accuracy.

For PDF malware classification, we compare *NU-$\delta$-MDt* with *Uniform-$\delta$* against EvadeML. We observe the best performances at $\|\delta\|_2 = 1$ for both methods. Table 1 depicts the clean accuracy (Ac.), AUC score and defense success rate (D.S.R.) against EvadeML for standard training, *Uniform-$\delta$* and *NU-$\delta$-MDt*. Although *NU-$\delta$-MDt* is a feature space defense, the results show that it is highly effective against problem space attacks, and it outperforms *Uniform-$\delta$*. Since our approach does not assume any attack knowledge, it is more generalizable than the problem space defenses.

Table 1: Clean accuracy (Ac.), AUC score and defense success rate (D.S.R.) against EvadeML for standard training, *Uniform-$\delta$* and *NU-$\delta$-MDt*.

| Model | $\|\delta\|_2$ | Clean Ac. | AUC score | D.S.R. |
|---|---|---|---|---|
| Std. training | - | 99.52% | 0.99912 | 11.1% |
| Uniform-$\delta$ | 1 | 97.64% | 0.99826 | 87.4% |
| NU-$\delta$-MDt | 1 | 97.83% | 0.99835 | **92.9%** |

## 3.2 Credit Risk Use-case

Our second use-case is a credit risk detection problem where the DNN's goal is to make decisions on loan applications for bank customers. For this scenario, we use the well-known German Credit dataset [40], which contains classes "good" and "bad", as well as applicant features such as age, employment status, income, savings, etc. It has 20 features and 1000 samples with 300 in the "bad" class. Similar to [17], we treat discrete features as continuous and drop non-ordinal categorical features.

**Attacks used for evaluation:** The goal of an adversary in this situation is to make DNN models conclude that they are approved for a loan when they actually may not be eligible. Since modifications to tabular data can be detected by an expert eye, attackers try to fool classifiers with imperceptible attacks. We use German Credit dataset implementation of LowProFool [17] which considers attack imperceptibility and represents expert knowledge using feature correlations. We apply the attack on the "bad" class of the test set and generate 155 AEs. After dropping the non-ordinal categorical features, we treat the remaining 12 features as continuous values.

**Numeric results:** Similar to the malware use-case, $\epsilon$ for each method is selected such that their average $\|\delta\|_2$ are approximately equal. In Figure 1b, we report defense success rate of PGD with

Table 2: Clean accuracy (Cl. Ac.) and defense success rates of *NU-δ-MDt* and *Uniform-δ* against FGSM, Carlini-Wagner (CW), JSMA and Deep Fool attacks for Spam Detection Use-case.

| Defenses | $\delta$ | Cl. Ac.,% | FGSM $\epsilon$=0.5,% | FGSM $\epsilon$=1,% | CW,% | JSMA,% | Deep Fool,% |
|---|---|---|---|---|---|---|---|
| Uniform-$\delta$ | 0.1 | 91.61 | $24.1 \pm 0.25$ | $20.17 \pm 0.31$ | $38.93 \pm 0.54$ | $41.51 \pm 0.24$ | $39.3 \pm 0.27$ |
| NU-$\delta$-MDt | | 91.93 | $\mathbf{28.7 \pm 0.17}$ | $\mathbf{27.1 \pm 0.21}$ | $\mathbf{49.62 \pm 0.33}$ | $\mathbf{49.59 \pm 0.18}$ | $\mathbf{51.73 \pm 0.31}$ |
| Uniform-$\delta$ | 0.3 | 90.40 | $25.22 \pm 0.12$ | $21.61 \pm 0.43$ | $45.83 \pm 0.41$ | $46.38 \pm 0.27$ | $47.39 \pm 0.25$ |
| NU-$\delta$-MDt | | 91.31 | $\mathbf{32.15 \pm 0.34}$ | $\mathbf{30.88 \pm 0.36}$ | $\mathbf{53.45 \pm 0.22}$ | $\mathbf{52.68 \pm 0.13}$ | $\mathbf{54.61 \pm 0.19}$ |
| Uniform-$\delta$ | 0.5 | 87.12 | $27.05 \pm 0.50$ | $23.08 \pm 0.34$ | $50.05 \pm 0.32$ | $49.5 \pm 0.25$ | $49.75 \pm 0.16$ |
| NU-$\delta$-MDt | | 87.78 | $\mathbf{43.85 \pm 0.62}$ | $\mathbf{36.12 \pm 0.20}$ | $\mathbf{55.24 \pm 0.38}$ | $\mathbf{53.71 \pm 0.41}$ | $\mathbf{64.28 \pm 0.55}$ |
| Uniform-$\delta$ | 1 | 86.46 | $40.20 \pm 0.57$ | $32.98 \pm 0.31$ | $52.77 \pm 0.24$ | $52.94 \pm 0.26$ | $53.22 \pm 0.10$ |
| NU-$\delta$-MDt | | 87.64 | $\mathbf{81.34 \pm 0.83}$ | $\mathbf{79.85 \pm 0.75}$ | $\mathbf{61.89 \pm 0.37}$ | $\mathbf{72.93 \pm 0.77}$ | $\mathbf{88.75 \pm 0.78}$ |
| Uniform-$\delta$ | 1.5 | 85.98 | $74.40 \pm 0.47$ | $62.36 \pm 0.75$ | $59.71 \pm 0.28$ | $68.95 \pm 0.85$ | $64.5 \pm 0.86$ |
| NU-$\delta$-MDt | | 87.03 | $\mathbf{94.45 \pm 0.18}$ | $\mathbf{91.03 \pm 0.10}$ | $\mathbf{72.98 \pm 0.48}$ | $\mathbf{86.43 \pm 0.32}$ | $\mathbf{98.36 \pm 0.25}$ |

uniform and non-uniform perturbations in detecting 155 AEs generated by LowProFool. The figure shows that for every given $\|\delta\|_2$, non-uniform perturbations outperform uniform perturbations in PGD. Although LowProFool represents feature importance by Pearson correlation coefficients between features and the output score, surprisingly *NU-δ-Pearson* is the best approach among the other non-uniform approaches for only $\delta = \{0.7, 1\}$. We refer to Table A.3 for clean accuracy results.

### 3.3 Spam Detection Use-case

Finally, we evaluate robustness within the context of detecting spam within social networks. We use a dataset from Twitter, where data from legitimate users and spammers is harvested from social honeypots over seven months [41]. This dataset contains profile information and posts of both spammers and legitimate users. After pre-processing [42], we extract 31 numeric features with 14 being integers and the rest being continuous. Some examples of these features are the number of following and followers as well as the ratio between them, percentage of bidirectional friends, number of posted messages per day, etc. We treat all features as continuous values in our experiments. Moreover, we extract 41,354 samples where the training set has 17,744 "bad" and 15,339 "good" samples, and the testing set has 3885 "bad" and 4386 "good" samples. The adversary's goal is to make the DNN predict that a tweet was posted by a legitimate user when it was written by a spammer.

**Attacks used for evaluation:** We incorporate the evasion attack [43] from [16] for our Twitter spam detector. The attack strategy is based on minimizing the maliciousness score of an AE which is measured by a local interpretation model LASSO, while satisfying $\ell_2$ norm constraint on perturbations. We generate the AEs by constraining the perturbations to $0.5 \times dist_{pos-neg}^{avg}$, where $dist_{pos-neg}^{avg}$ is defined by [16] as the average distance between the spammer samples and the closest non-spammers to these samples. We split the Twitter dataset with ratio 25% for training and testing, and generate the AEs using the spammer class of the entire test set.

**Numeric results:** Again, we apply perturbations only to the spammer set during AT and report the results for approximately equal average $\|\delta\|_2$ perturbations. Figure 1c illustrates defense success rate in detecting AEs of the proposed approaches against the model interpretation based attack [16] for Twitter dataset. The figure shows that non-uniform perturbations outperform uniform case in terms of defense S.R. for all given $\|\delta\|_2$. We refer to Table A.4 for clean accuracy results.

### 3.4 Performance Against Uniform Attacks

Throughout the experiments, we tested our non-uniform approach against various realistic attacks, such as problem space attacks in Section 3.1, feature importance-based attack in Section 3.2 and explainability-based attack in Section 3.3. So far we emphasized that problem space attacks and non-uniformly norm bounded attacks are more realistic compared to the traditional uniformly norm-bounded attacks which are mostly considered in image domain. Yet, in this section, we also test our non-uniform approach against the well-known uniform attacks to investigate the generalizability of our approach. We use the setting for the spam detection use-case, and compare *NU-δ-MDt* with *Uniform-δ*. We utilize the adversarial robustness toolbox (ART) [44] to craft AEs by using the default parameters for the AE generators of Carlini-Wagner (CW), JSMA and DeepFool Methods. We also

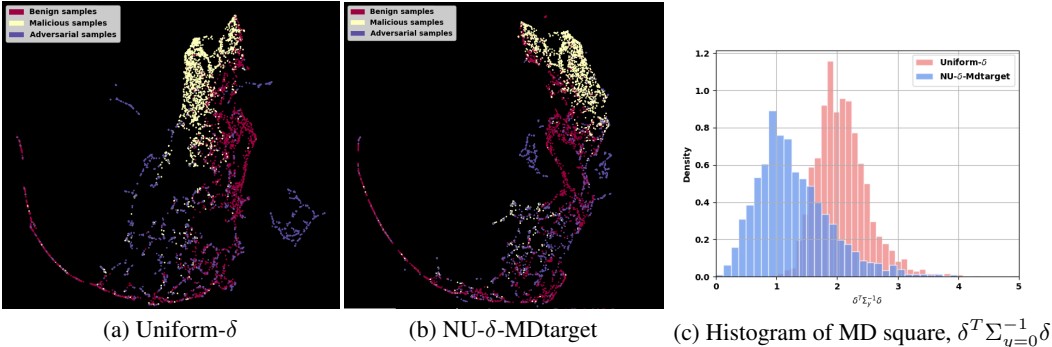

(a) Uniform-$\delta$    (b) NU-$\delta$-MDtarget    (c) Histogram of MD square, $\delta^T \Sigma_{y=0}^{-1} \delta$

Figure 2: UMAP visualization of benign, malicious and adversarial samples generated by (a) Uniform-$\delta$ and (b) NU-$\delta$-MDtarget, and (c) the density histogram of their $\delta^T \Sigma_{y=0}^{-1} \delta = 2C - 2 \log \gamma$.

use FGSM for $\epsilon = 0.5$ and $\epsilon = 1$. Table 2 shows the clean accuracy (Cl. Ac.) and defense S.R.'s of the robust models *NU-$\delta$-MDt* and *Uniform-$\delta$*. We observe in Table 2 that the Cl. Ac. decreases as $||\delta||_2$ increases for both *Uniform-$\delta$* and *NU-$\delta$-MDt* but the degradation in non-uniform is less. Defense S.R.'s, on the other hand, improve for both approaches but *NU-$\delta$-MDt* significantly outperforms *Uniform-$\delta$* in all cases. See Appendix A.7 for the performance against uniform PGD attacks.

### 3.5 Quality of Perturbation Sets

In this section, we quantitatively and qualitatively analyze how well non-uniform perturbations capture realistic attacks using $\gamma$-consistency property defined in Section 2 and lower dimensional space visualization. Our intuition is that a successful attack evades detection since AEs appear benign to the model. That is, AEs have high likelihood according to the distribution of benign samples. Therefore, we measure a perturbed sample's quality by its $\gamma$-consistency with the benign set distribution. Definition 2.1 leverages Theorem 2.1, which shows that smaller MD for $\delta$ indicates higher $\gamma$-consistency and hence higher quality of the perturbed sample. Moreover, we expect AEs that evade the model and benign samples to be embedded closer to each other in the lower-dimensional subspace. Figure 2 illustrates UMAP visualization [45] of benign, malicious and adversarial samples for the spam detection use-case. AEs generated by NU-$\delta$-MDtarget show better alignment with benign distribution, which shows that NU-$\delta$-MDtarget mimics a more realistic attack. We also show the histogram of MD squares, i.e. $\delta^T \Sigma_{y=0}^{-1} \delta = 2C - 2 \log \gamma$, of 1660 AEs from Uniform-$\delta$ and NU-$\delta$-MDtarget in Figure 2c, where the average values are 2.1 and 1.28, respectively. Following Theorem 2.1 and Figure 2c, $\delta$'s from NU-$\delta$-MDtarget have higher $\gamma$, and hence, are more realistic.

## 4 Certified Robustness with Non-uniform Perturbations

In this section, we present methods for certifying robustness with non-uniform perturbations. We consider two well-known methods; linear programming (LP) [9] and randomized smoothing [46].

### 4.1 LP Formulation

We can provably certify the robustness of deep ReLU networks against non-uniform adversarial perturbations at the input. Our derivation follows an LP formulation of the adversary's problem with ReLU relaxations, then the dual problem of the LP and activation bound calculation. It can be viewed as an extension of [9]. Similar to [9], we consider a $k$ layer feedforward deep ReLU network with

$$\hat{z}_{i+1} = W_i z_i + b_i, \ z_i = \max\{\hat{z}_i, 0\}, \text{ for } i = 1, \cdots, k-1 \tag{7}$$

We denote $\mathcal{Z}_{\epsilon,\Omega}(x) := \{f_\theta(x + \delta) : ||\Omega\delta||_p \leq \epsilon\}$ as the set of all attainable final-layer activations by input perturbation $\delta$. Since this is a non-convex set for multi-layer networks which is hard to optimize over, we consider a convex outer bound on $\mathcal{Z}_{\epsilon,\Omega}(x)$ and optimize the worst case loss over this bound to guarantee that no AEs within $\mathcal{Z}_{\epsilon,\Omega}(x)$ can evade the network. As done in [9], we relax the ReLU activations by representing $z = \max\{0, \hat{z}\}$ with their upper convex envelopes

$z \geq 0, z \geq \hat{z}, -u\hat{z} + (u - l)z \leq -ul$, where $l$ and $u$ are the known lower and upper bounds for the pre-ReLU activations. We denote the new relaxed set of all attainable final-layer activations by $\tilde{\mathcal{Z}}_{\epsilon,\Omega}(x)$. Assuming that an adversary targets a specific class to fool the classifier, we write the LP as

$$\underset{\hat{z}_k}{\text{minimize}} \quad c^T \hat{z}_k \qquad \text{s.t.} \quad \hat{z}_k \in \tilde{\mathcal{Z}}_{\epsilon,\Omega} \tag{8}$$

where $c := e_{y^{true}} - e_{y^{target}}$ is the difference between the selection vector of true and the target class.

A positive valued objective for all classes as a solution to equation 8 indicates that there is no adversarial perturbation within $\tilde{\Delta}_{\epsilon,p}$ which can evade the classifier. To be able to solve equation 8 in a tractable way, we consider its dual whose feasible solution provides a guaranteed lower bound for the LP. It is previously shown by [9] that a feasible set of the dual problem can be formulated similar to a standard backpropagation network and solved efficiently. The dual problem of our LP with ReLU relaxation and non-uniform perturbation constraints is expressed in the following theorem.

**Theorem 4.1.** *The dual of the linear program 8 can be written as*

$$\underset{\hat{\nu},\nu}{\text{maximize}} \quad -\sum_{i=1}^{k-1} \nu_{i+1}^T b_i + \sum_{i=2}^{k-1} \sum_{j \in \mathcal{I}_i} l_{i,j}[\hat{\nu}_{i,j}]_+ - \hat{\nu}_1^T x - \epsilon ||\Omega^{-1}\hat{\nu}_1||_q$$

$$\text{s.t.} \qquad \nu_k = -c, \ \hat{\nu}_i = (W_i^T \nu_{i+1}), \ \text{for} \ i = k-1,\dots,1 \tag{9}$$

$$\nu_{i,j} = \begin{cases} 0 & j \in \mathcal{I}_i^- \\ \hat{\nu}_{i,j} & j \in \mathcal{I}_i^+ \\ \frac{u_{i,j}}{u_{i,j}-l_{i,j}}[\hat{\nu}_{i,j}]_+ - \eta_{i,j}[\hat{\nu}_{i,j}]_- & j \in \mathcal{I}_i \end{cases}, \ \text{for} \ i = k-1,\dots,2$$

*where $\mathcal{I}_i^-$, $\mathcal{I}_i^+$ and $\mathcal{I}_i$ represent the activation sets in layer $i$ for $l$ and $u$ are both negative, both positive and span zero, respectively.*

See Appendix A.4 for the proof of Theorem 4.1. When $\eta_{i,j} = \frac{u_{i,j}}{u_{i,j}-l_{i,j}}$, Theorem 4.1 shows that the dual problem can be represented as a linear back propagation network, which provides a tractable solution for a lower bound of the primal objective. To solve equation 9, we need to calculate lower and upper bounds for each layer incrementally as explained in Appendix A.5.

For certification of robustness within a non-uniform norm ball around a test sample, we need the objective of the LP to be positive for all classes. Since the solution of the dual problem is a lower bound on the primal LP, it provides a worst case certification guarantee against the AEs within the non-uniform norm ball. We provide certification results for the robustness of *Uniform-δ* and *NU-δ-MDt* (*NU-δ-MDtarget*) for spam detection use-case in Table 3. We consider both uniform and non-uniform input constraints in certification, namely *Uniform-Cert* for the standard LP approach for certi-

Table 3: Average certification margin and number of successful certified samples out of 1000 spammers for *NU-δ-MDt* and *Uniform-δ* for Spam Detection Use-case.

| Model | Defense S.R. | Cert. Method | Margin | Cert. Success |
|---|---|---|---|---|
| Uniform-δ | $54.87 \pm 1.1\%$ | Uniform-Cert | 1.07 | $34.72 \pm 0.94\%$ |
| | | NU-Cert-SHAP | 1.84 | $72.64 \pm 0.6\%$ |
| | | NU-Cert-Pearson | 2.04 | $76.8 \pm 0.71\%$ |
| | | NU-Cert-MD | 2.40 | $80.2 \pm 0.56\%$ |
| | | NU-Cert-MDt | 2.40 | $80.2 \pm 0.55\%$ |
| NU-δ-MDt | $63.4 \pm 0.74\%$ | Uniform-Cert | 1.11 | $42.95 \pm 0.69\%$ |
| | | NU-Cert-SHAP | 1.9 | $74.65 \pm 0.85\%$ |
| | | NU-Cert-Pearson | 2.06 | $78.38 \pm 0.76\%$ |
| | | NU-Cert-MD | 2.41 | $81.3 \pm 0.68\%$ |
| | | NU-Cert-MDt | 2.41 | $81.3 \pm 0.67\%$ |

fication with uniform perturbation constraint [9], and *NU-Cert-(.)* for the non-uniform constraint. We implement our non-uniform approach into the LP by modifying [9] with our $\Omega$ matrix, and generate various certification methods by non-uniform $\Omega$, e.g. *NU-Cert-SHAP*, *NU-Cert-Pearson*, *NU-Cert-MD* and *NU-Cert-MDt*. Our purpose is not to propose the tightest certification bounds but to show that non-uniform constraints result in larger certification margins compared to the uniform case.

We compare *Uniform-δ* and *NU-δ-MDt* to evaluate certification results. Dropout layers are removed from the model for LP solution, and AT is performed for $\epsilon = 0.3$. Certification is done by solving the LP for $\epsilon = 0.3$ over 1000 spammers. The objective should be positive for all classes to certify the corresponding sample. The margin between the objective and zero gives an idea about how

tight the bound is [47]. Table 3 demonstrates two main results: (i) the certification success of *NU-δ-MDtarget* over *Uniform-δ* for each certification method supports our claim that non-uniform perturbations provide higher robustness than the uniform approach; and (ii) certification with non-uniform constraints provide larger certification margins and hence tighter bound.

## 4.2 Randomized Smoothing

Robustness certification via *randomized smoothing* [46] is an empirical alternative to the LP. The idea is constructing a "smoothed" classifier $g$ from the base classifier $f$. In the original formulation in [46], $g$ returns the most likely output returned by $f$ given input $x$ is perturbed by isotropic Gaussian noise. Here, we provide robustness guarantee in binary case for randomized smoothing framework when non-isotropic Gaussian noise is used to allow robustness to non-uniform perturbations:

$$g(x) = \arg\max_{y \in \mathcal{Y}} \mathbb{P}(f(x+n) = y) \quad \text{where} \quad n \sim \mathcal{N}(0, \Sigma). \tag{10}$$

Adapting notation and Theorem 2 from [46], let $p_a$ be the probability of the most probable class $y = a$ when the base classifier $f$ classifies $\mathcal{N}(x, \Sigma)$. Then the below theorem holds.

**Theorem 4.2.** *In binary classification problem, suppose $\underline{p_a} \in (\frac{1}{2}, 1]$ satisfies $\mathbb{P}(f(x+n) = a) \geq \underline{p_a}$. Then $g(x + \delta) = a$ for all $\sqrt{\delta^T \Sigma^{-1} \delta} \leq \Phi_{r,n}^{-1}(\underline{p_a}) - q_{50}$, where $r := \sqrt{\delta^T \Sigma^{-1} \delta}$, $\Phi_{r,n}^{-1}(\underline{p_a})$ is the quantile function of the $\chi$ distribution of $d$ degrees of freedom, and $q_{50}$ is the $50^{th}$ quantile.*

See Appendix A.4 for the proof. In theorem 4.2, we show that a smoothed classifier $g$ is robust around $x$ within $\ell_2$ Mahalanobis distance $\sqrt{\delta^T \Sigma^{-1} \delta} \leq \Phi_{r,n}^{-1}(p_a) - q_{50}$, where $\Phi_{r,n}^{-1}(p_a)$ is the quantile function for probability $p_a$. The same result holds if we replace $p_a$ with lower bound $\underline{p_a}$.

We implement our non-uniform approach into randomized smoothing by modifying [48] with our non-isotropic noise space. Table 4 shows certification S.R. of Uniform-δ and NU-δ-MDt, when they are certified by standard randomized smoothing with $\mathcal{N}(0, \sigma I)$ (UC), and our non-uniform methods with $\mathcal{N}(0, \Sigma_y)$

Table 4: Percentage of successfully certified samples for *NU-δ-MDt* and *Uniform-δ* with various certification approaches with randomized smoothing for Spam Detection use-case.

| Model | UC | NUC-Pearson | NUC-SHAP | NUC-MD | NUC-MDt |
|---|---|---|---|---|---|
| Uniform-δ | 50.96% | 61.11% | 64.24% | 65.72% | 66.45% |
| NU-δ-MDt | **61.8%** | **67.14%** | **71.25%** | **85.34%** | **90.11%** |

for corresponding $\Sigma_y$. That is, $\Sigma_{y=0}$ for NUC-MDt, $\Sigma_{y=\{0,1\}}$ NUC-MD, $\frac{1}{\rho^2} I$ for NUC-Pearson and $\frac{1}{s^2} I$ for NUC-SHAP are used when the average training distortion budget is $\|\delta\|_2 = 5$ and the average certification distortion is $\|\delta\|_2 = 2.8$. Table 4 shows that NU-δ-MDt is certifiably robust for more samples than Uniform-δ for all certification methods. Moreover, certification with non-uniform noise, especially with NUC-MDt, provides higher certification S.R. compared to uniform noise.

## 5 Conclusion

In this work, we study adversarial robustness against evasion attacks, with a focus on applications where input features have to comply with certain domain constraints. We assume Gaussian data distribution in our consistency analysis, as well as precomputed covariance matrix and Shapley values. Under these assumptions, our results on three different applications demonstrate that non-uniform perturbation sets in AT improve adversarial robustness, and non-uniform bounds provide better robustness certification. As an unintended negative social impact, our insights might be used by malicious parties to generate AEs. However, this work provides the necessary defense mechanisms against these potential attacks.

## Acknowledgments and Disclosure of Funding

We would like to thank Mohamad Ali Torkamani for his help and valuable feedback, and thank Baris Coskun for his endless support as the leader of the Security Analytics and AI Research (SAAR) Team at AWS, as well as the team members for the fruitful discussions. This work has been done as a part of Ecenaz Erdemir's internship in AWS, and no outside funding was used in the preparation.

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
