# Supplementary Material for Adversarial Robustness with Non-uniform Perturbations

**Ecenaz Erdemir** *
Imperial College London
e.erdemir17@imperial.ac.uk

**Jeffrey Bickford**
Amazon Web Services
jbick@amazon.com

**Luca Melis**
Amazon Web Services
lucmeli@amazon.com

**Sergül Aydöre**
Amazon Web Services
saydore@amazon.com

## A.1 Toy Example for Non-uniform Perturbations

Adversarial training can be represented as a *min-max optimization*. Given a dataset $\{x_i, y_i\}_{i=1}^n$ with input $x_i \in \mathbb{R}^d$ and classes $y_i \in \mathcal{Y}$, the objective of adversarial training is denoted by

$$\min_\theta \frac{1}{n} \sum_{i=1}^n \max_{\delta \in \Delta} \ell(f_\theta(x_i + \delta), y_i) \tag{A.1}$$

where $f_\theta : \mathbb{R}^d \to \mathcal{Y}$ is DNN function, $\ell(.)$ is the loss, e.g. cross-entropy, and $\Delta$ is the set of possible adversarial perturbations around the original samples. The adversary's objective is the inner maximization term in equation A.1, and the perturbed samples found as a solution to the norm-constrained inner maximization are called adversarial examples, or AEs.

Consider the 2D toy example of binary classification in Figure A.1 which is obtained by modifying [1]. Figure A.1 illustrates adversarially robust decision boundaries with red and blue regions, and $l_2$-norm perturbation limits around the data points with black circles. While Figure A.1a shows that adversarially trained model with input constraint $\|\delta\|_2 \leq 0.5$ gains complete robustness against input perturbations, in Figure A.1b there is loss of clean performance due to overlapping regions of increased allowed perturbations. Although the constraint $\|\delta\|_2 \leq 0.5$ might provide sufficient robustness in $x$-axis, there are still uncovered regions in $y$-axis in Figure A.1a. On the other hand, when we fit the allowable perturbations to $y$-axis by choosing a larger perturbation $\|\delta\|_2 \leq 0.8$, $x$-axis suffers from unnecessary overlaps. This can be solved by customizing the perturbation constraint such that the perturbation radius in x-axis follows $|\delta_x| \leq 0.5$ and the radius in y-axis follows $|\delta_y| \leq 0.8$, which results in an ellipsoid perturbation

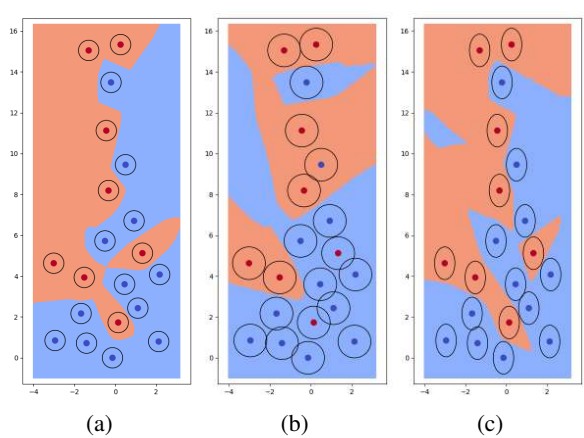

Figure A.1: Classification boundaries from adversarial training with uniform perturbation limits for (a) $\|\delta\|_2 \leq 0.5$, (b) $\|\delta\|_2 \leq 0.8$ and non-uniform perturbation limits for (c) $|\delta_x| \leq 0.5$ and $|\delta_y| \leq 0.8$.

---

*Work done at Amazon Web Services.

35th Conference on Neural Information Processing Systems (NeurIPS 2021).

region in 2D as shown in Figure A.1c. This toy example highlights the advantage of a non-uniform constraint across both axes.

Uniformly perturbing all pixels in an image is often imperceptible to the human eye, but uniform perturbations are wholly inappropriate in many tabular datasets, where positive and negative correlations are strong, consistent, and meaningful. For example, in the German dataset used in Section 3.2, we find the largest positive correlation (0.62) between the amount of credit and the payment duration, while the largest negative correlation (-0.31) is between the checking account status and the credit risk score. Both relationships are intuitive, and both would be broken by applying uniform perturbations.

## A.2 Ember Dataset Features

The EMBER dataset [2] consists of two types of features:

1. **Parsed features** are extracted after parsing the portable executable (PE) file. Parsed features include 5 different groups:
   - *General file information*: virtual size of the file; number of imported/exported functions and symbols; whether the file has a debug section, thread local storage, resources, relocations, or a signature.
   - *Header information*: timestamp in the header; target machine; list of image and DLL characteristics; target subsystem; file magic; image, system and subsystem versions; code, headers and commit sizes (hashing trick).
   - *Imported functions*: functions extracted from the import address table (hashing trick)
   - *Exported functions*: list of exported functions (hashing trick).
   - *Section information*: name, size, entropy, virtual size, and a list of strings representing section characteristics (hashing trick).
2. **Format-agnostic features** do not require parsing the PE file structure and include:
   - *Byte histogram*: counts of each byte value within the file (256 integer values).
   - *Byte-entropy histogram*: quantized and normalized of the joint distribution $p(H, X)$ of entropy $H$ and byte value $X$ (256 bins).
   - *String information*: number of strings and their average length; a histogram of the printable characters within those strings; entropy of characters across all printable strings.

## A.3 DNN Architecture and Pre-processing

In all applications, we use a fully-connected neural network model composed of 4 densely connected layers with the first three using *ReLU* activations followed by a *softmax* activation in the last layer. After each of the first three layers, we apply $20\%$ *Dropout* rate for regularization during training. We use 5 random initialization for malware and 10 for both credit risk and spam detection use-cases to report average results.

For pre-processing, we use standardization as a normalization method, which is a common practice with many machine learning techniques. Min-max scaling transforms all features into the same scale while standardization, which is recommended in presence of outliers [3], only ensures zero mean and unit standard deviation. This approach does not guarantee same range (min and max) for all features. As a result, it is possible that the features have different scales even after normalization.

## A.4 Theorem Proofs

**Theorem 3.1.** *If the AEs are generated according to MD constraint, then their $\gamma$-consistency has a direct relation to $\epsilon$ such that*

$$0 < \sqrt{2C - 2\log\gamma} \leq \epsilon. \tag{A.2}$$

*where $C = -\log(2\pi)^{d/2}|\Sigma_y|^{1/2}$, $d$ is the dimension of $x$, and $\sqrt{\delta^T \Sigma_y^{-1} \delta} \leq \epsilon$.*

*Proof.* For an AE $x$ that is generated under the Mahalanobis distance constraint, i.e., $x \in \tilde{\Delta}_{\epsilon,2}$, we can write the following bound:

$$\min_{x \in \tilde{\Delta}_{\epsilon,2}} \log f(x \mid y) = C - \frac{1}{2}\delta^T \Sigma_y^{-1} \delta = \log \gamma \tag{A.3}$$

where the second equality is a result of $\gamma$-consistency assumption. Then, by using the upper limit of $\ell_2$ Mahalanobis distance of $\delta$ for $M = \Sigma_y^{-1}$, we get

$$\sqrt{\delta^T \Sigma_y^{-1} \delta} = \sqrt{2C - 2\log\gamma} \le \epsilon. \tag{A.4}$$

$\square$

Theorem 2.1 implies that there is a direct relationship between limiting the Mahalonobis distance of $\delta$ and ensuring consistent samples when the data is Gaussian.

**Theorem 4.1.** *The dual of the linear program 8 can be written as*

$$\begin{aligned}
\underset{\hat{\nu},\nu}{maximize} \quad & -\sum_{i=1}^{k-1} \nu_{i+1}^T b_i + \sum_{i=2}^{k-1}\sum_{j\in\mathcal{I}_i} l_{i,j}[\hat{\nu}_{i,j}]_+ - \hat{\nu}_1^T x - \epsilon\|\Omega^{-1}\hat{\nu}_1\|_q \\
s.t. \quad & \nu_k = -c, \ \hat{\nu}_i = (W_i^T \nu_{i+1}), \ \ for \ \ i = k-1,\dots,1 \\
& \nu_{i,j} = \begin{cases} 0 & j \in \mathcal{I}_i^- \\ \hat{\nu}_{i,j} & j \in \mathcal{I}_i^+ \ , \ \ for \ \ i = k-1,\dots,2 \\ \frac{u_{i,j}}{u_{i,j}-l_{i,j}}[\hat{\nu}_{i,j}]_+ - \eta_{i,j}[\hat{\nu}_{i,j}]_- & j \in \mathcal{I}_i \end{cases}
\end{aligned} \tag{A.5}$$

*where $\mathcal{I}_i^-$, $\mathcal{I}_i^+$ and $\mathcal{I}_i$ represent the activation sets in layer $i$ for $l$ and $u$ are both negative, both positive and span zero, respectively.*

*Proof.* The linear program with non-uniform input perturbation and relaxed ReLU constraints can be written as

$$\begin{aligned}
\underset{\hat{z}_k}{minimize} \quad & c^T \hat{z}_k \\
s.t. \quad & \hat{z}_{i+1} = W_i z_i + b_i, i = 1,\dots,k-1 \\
& \|\Omega(z_1 - x)\|_p \le \epsilon \\
& z_{i,j} = 0, i = 2,\dots,k-1, j \in \mathcal{I}_i^- \\
& z_{i,j} = \hat{z}_{i,j}, i = 2,\dots,k-1, j \in \mathcal{I}_i^+ \\
& \left.\begin{aligned} z_{i,j} \ge 0, \ z_{i,j} \ge \hat{z}_{i,j}, \\ ((u_{i,j}-l_{i,j})z_{i,j} - u_{i,j}\hat{z}_{i,j}) \le -u_{i,j}l_{i,j} \end{aligned}\right\} \begin{aligned} i=2,\dots,k-1 \\ j\in\mathcal{I}_i \end{aligned}.
\end{aligned} \tag{A.6}$$

We associate the following Lagrangian variables with each of the constraints except the $\ell_p$ norm constraint in Problem A.6,

$$\begin{aligned}
\hat{z}_{i+1} = W_i z_i + b_i &\Rightarrow \nu_{i+1} \\
\delta = z_1 - x &\Rightarrow \psi \\
-z_{i,j} \le 0 &\Rightarrow \mu_{i,j} \\
\hat{z}_{i,j} - z_{i,j} \le 0 &\Rightarrow \tau_{i,j} \\
((u_{i,j}-l_{i,j})z_{i,j} - u_{i,j}\hat{z}_{i,j}) \le -u_{i,j}l_{i,j} &\Rightarrow \lambda_{i,j}.
\end{aligned} \tag{A.7}$$

We do not define explicit dual variables for $z_{i,j} = 0$ and $z_{i,j} = \hat{z}_{i,j}$ since they will be zero in the optimization. Then, we create the following Lagrangian by grouping up the terms with $z_i, \hat{z}_i$:

$$L(\mathbf{z}, \hat{\mathbf{z}}, \nu, \delta, \lambda, \tau, \mu, \psi) = -(W_1^T \nu_2 + \psi)^T z_1 - \sum_{\substack{i=2 \\ j \in \mathcal{I}_i}}^{k-1} (\mu_{i,j} + \tau_{i,j} - \lambda_{i,j}(u_{i,j} - l_{i,j}) + (W_i^T \nu_{i+1})_j) z_{i,j}$$

$$+ \sum_{\substack{i=2 \\ j \in \mathcal{I}_i}}^{k-1} (\tau_{i,j} - \lambda_{i,j} u_{i,j} + \nu_{i,j}) \hat{z}_{i,j} + (c + \nu_k)^T \hat{z}_k - \sum_{i=1}^{k-1} \nu_{i+1}^T b_i$$

$$+ \sum_{\substack{i=2 \\ j \in \mathcal{I}_i}}^{k-1} \lambda_{i,j} u_{i,j} l_{i,j} + \psi^T x + \psi^T \delta$$

$$\text{subject to} \quad ||\Omega \delta||_p \le \epsilon$$

(A.8)

Now, we take the minimum of $L(.)$ w.r.t $\mathbf{z}, \hat{\mathbf{z}}$ and $\delta$:

$$\inf_{\mathbf{z}, \hat{\mathbf{z}}, \delta} L(\mathbf{z}, \hat{\mathbf{z}}, \nu, \delta, \lambda, \tau, \mu, \psi) = -\inf_{z_{i,j}} \sum_{\substack{i=2 \\ j \in \mathcal{I}_i}}^{k-1} \left( \mu_{i,j} + \tau_{i,j} - \lambda_{i,j}(u_{i,j} - l_{i,j}) + (W_i^T \nu_{i+1})_j \right) z_{i,j}$$

$$+ \inf_{\hat{\mathbf{z}}} \Big( \sum_{\substack{i=2 \\ j \in \mathcal{I}_i}}^{k-1} (\tau_{i,j} - \lambda_{i,j} u_{i,j} + \nu_{i,j}) \hat{z}_{i,j} + (c + \nu_k)^T \hat{z}_k \Big) - \sum_{i=1}^{k-1} \nu_{i+1}^T b_i$$

$$+ \sum_{\substack{i=2 \\ j \in \mathcal{I}_i}}^{k-1} \lambda_{i,j} u_{i,j} l_{i,j} + \psi^T x + \inf_{||\Omega \delta||_p \le \epsilon} \psi^T \delta - \inf_{z_1} (W_1^T \nu_2 + \psi)^T z_1.$$

(A.9)

We can represent the term $\inf_{||\Omega \delta||_p \le \epsilon} \psi^T \delta$ independent of $\delta$ using the following dual norm definition.

**Cauchy-Schwarz inequality for dual norm:**

We can write the Cauchy-Schwarz inequality as $\alpha^T \beta \le ||\alpha||_p ||\beta||_q$, where $\frac{1}{p} + \frac{1}{q} = 1$ and $q$ norm represents the dual of $p$ norm. Let $\hat{u} = \frac{\alpha}{||\alpha||_p}$, the definition of dual norm is

$$||\beta||_q = \sup_{||\hat{u}||_p \le 1} \hat{u}^T \beta.$$

(A.10)

We can write $\inf_{||\Omega \delta||_p \le \epsilon} \psi^T \delta = -\sup_{||\Omega \delta||_p \le \epsilon} (-\psi^T \delta) = -\sup_{||\Omega \delta||_p \le \epsilon} \psi^T \delta$. For $\alpha = \frac{\Omega \delta}{\epsilon}$ and $\beta = \epsilon \Omega^{-1} \psi$, we get $\delta^T \psi \le ||\frac{\Omega \delta}{\epsilon}||_p ||\epsilon \Omega^{-1} \psi||_q$ which implies $-\sup_{||\Omega \delta||_p \le \epsilon} \psi^T \delta = -\epsilon ||\Omega^{-1} \psi||_q$.

Hence, the minimization of $L(.)$ becomes,

$$\inf_{\mathbf{z}, \hat{\mathbf{z}}, \delta} L(.) = \begin{cases} -\sum_{i=1}^{k-1} \nu_{i+1}^T b_i + \sum_{\substack{i=2 \\ j \in \mathcal{I}_i}}^{k-1} \lambda_{i,j} u_{i,j} l_{i,j} + \psi^T x - \epsilon ||\Omega^{-1} \psi||_q & \text{if } cond. \\ \\ -\infty & \text{o.w.,} \end{cases}$$

(A.11)

where the conditions are

$$\nu_k = -c$$
$$W_1^T \nu_2 = -\psi$$
$$\nu_{i,j} = 0, \; j \in \mathcal{I}_i^-$$
$$\nu_{i,j} = (W_i^T \nu_{i+1})_j, \; j \in \mathcal{I}_i^+$$
$$\left. \begin{aligned} ((u_{i,j} - l_{i,j})\lambda_{i,j} - \mu_{i,j} - \tau_{i,j}) &= (W_i^T \nu_{i+1})_j \\ \nu_{i,j} &= u_{i,j} \lambda_{i,j} - \tau_{i,j} \end{aligned} \right\} \begin{smallmatrix} i=2,\ldots,k-1 \\ j \in \mathcal{I}_i \end{smallmatrix}.$$

(A.12)

The dual problem can be rearranged and reduced to the standard form

$$\underset{\nu,\psi,\lambda,\tau,\mu}{\text{maximize}} \quad -\sum_{i=1}^{k-1}\nu_{i+1}^T b_i + \psi^T x - \epsilon||\Omega^{-1}\psi||_q + \sum_{i=2}^{k-1}\lambda_i^T(u_i l_i) \tag{A.13}$$

$$\text{s.t.} \quad \nu_k = c \tag{A.14}$$

$$W_1^T \nu_2 = -\psi \tag{A.15}$$

$$\nu_{i,j} = 0, \; j \in \mathcal{I}_i^- \tag{A.16}$$

$$\nu_{i,j} = (W_i^T \nu_{i+1})_j, \; j \in \mathcal{I}_i^+ \tag{A.17}$$

$$\left.\begin{array}{l}((u_{i,j}-l_{i,j})\lambda_{i,j}-\mu_{i,j}-\tau_{i,j}) = (W_i^T\nu_{i+1})_j \\ \nu_{i,j} = u_{i,j}\lambda_{i,j}-\tau_{i,j}\end{array}\right\} \begin{array}{l} i=2,\ldots,k-1 \\ j \in \mathcal{I}_i\end{array} \tag{A.18}$$

$$\lambda, \tau, \mu \geq 0. \tag{A.19}$$

The insight of the dual problem is that it can also be written in the form of a deep network. Consider the equality constraint A.18, the dual variable $\lambda$ corresponds to the upper bounds in the convex ReLU relaxation, while $\mu$ and $\tau$ correspond to the lower bounds $z \geq 0$ and $z \geq \hat{z}$, respectively. By the complementary property, these variables will be zero of ReLU constraint is non-tight, and non-zero if the ReLU constraint is tight. since the upper and lower bounds cannot be tight simultaneously, either $\lambda$ or $\mu + \tau$ must be zero. Hence, at the optimal solution to the dual problem,

$$(u_{i,j}-l_{i,j})\lambda_{i,j} = [(W_i^T\nu i+1)_j]_+ \tag{A.20}$$
$$\tau_{i,j}+\mu_{i,j} = [(W_i^T\nu i+1)_j]_-.$$

Combining this with the constraint $\nu_{i,j} = u_{i,j}\lambda_{i,j}-\tau_{i,j}$ leads to

$$\nu_{i,j} = \frac{u_{i,j}}{u_{i,j}-l_{i,j}}[(W_i^T\nu i+1)_j]_+ - \eta[(W_i^T\nu i+1)_j]_- \tag{A.21}$$

for $j \in \mathcal{I}_i$ and $0 \leq \eta \leq 1$. This is a leaky ReLU operation with a slope of $\frac{u_{i,j}}{u_{i,j}-l_{i,j}}$ in the positive portion and and a negative slope $\eta$ between $0$ and $1$. Also note that from A.15 $-\psi$ denotes the pre-activation variable for the first layer. For the sake of simplicity, we use $\hat{\nu}_i$ to denote the pre-activation variable for layer $i$, then the objective of the dual problem becomes

$$\begin{aligned}S_{D_\epsilon}(x,\nu) &= -\sum_{i=1}^{k-1}\nu_{i+1}^T b_i + \sum_{i=2}^{k-1}\sum_{j\in\mathcal{I}_i}\frac{u_{i,j}l_{i,j}}{u_{i,j}-l_{i,j}}[\hat{\nu}_{i,j}]_+ - \hat{\nu}_1^T x - \epsilon||\Omega^{-1}\hat{\nu}_1||_q \\ &= -\sum_{i=1}^{k-1}\nu_{i+1}^T b_i + \sum_{i=2}^{k-1}\sum_{j\in\mathcal{I}_i}l_{i,j}[\hat{\nu}_{i,j}]_+ - \hat{\nu}_1^T x - \epsilon||\Omega^{-1}\hat{\nu}_1||_q\end{aligned} \tag{A.22}$$

Hence, the final form of the dual problem can be rewritten as a network with objective $S_{D_\epsilon}(x,\nu)$, input $-c$ and activations $\mathcal{I}$ as follows:

$$\underset{\hat{\nu},\nu}{\text{maximize}} \quad -\sum_{i=1}^{k-1}\nu_{i+1}^T b_i + \sum_{i=2}^{k-1}\sum_{j\in\mathcal{I}_i}l_{i,j}[\hat{\nu}_{i,j}]_+ - \hat{\nu}_1^T x - \epsilon||\Omega^{-1}\hat{\nu}_1||_q$$

$$\text{s.t.} \quad \nu_k = -c$$
$$\hat{\nu}_i = (W_i^T\nu_{i+1}), i = k-1,\ldots,1 \tag{A.23}$$
$$\nu_{i,j} = \begin{cases} 0 & j \in \mathcal{I}_i^- \\ \hat{\nu}_{i,j} & j \in \mathcal{I}_i^+ \\ \frac{u_{i,j}}{u_{i,j}-l_{i,j}}[\hat{\nu}_{i,j}]_+ - \eta[\hat{\nu}_{i,j}]_- & j \in \mathcal{I}_i \end{cases} \quad i = k-1,\ldots,2$$

$$\square$$

**Theorem 4.2.** *In binary classification problem, suppose $\underline{p_a} \in (\frac{1}{2}, 1]$ satisfies $\mathbb{P}(f(x+n) = a) \geq \underline{p_a}$. Then $g(x+\delta) = a$ for all $\sqrt{\delta^T \Sigma^{-1} \delta} \leq \Phi_{r,n}^{-1}(\underline{p_a}) - q_{50}$, where $r := \sqrt{\delta^T \Sigma^{-1} \delta}$, $\Phi_{r,n}^{-1}(\underline{p_a})$ is the quantile function of the $\chi$ distribution of $d$ degrees of freedom, and $q_{50}$ is the $50^{th}$ quantile.*

*Proof.* Let $X$ and $Y$ be random variables such that $X \sim \mathcal{N}(x, \Sigma)$ and $Y \sim \mathcal{N}(x+\delta, \Sigma)$. Next, we define the set $\mathcal{A} := \left\{ z \mid \delta^T \Sigma^{-1}(z-x) \leq \sqrt{\delta^T \Sigma^{-1} \delta} \Phi_{r,d}^{-1}(\underline{p_a}) \right\}$, where $r := \sqrt{\delta^T \Sigma^{-1} \delta}$ and $\Phi_{r,n}^{-1}(\underline{p_a})$ is the quantile function of the $\chi$ distribution of $d$ degree of freedom for the probability $p_a$, so that $\mathbb{P}(X \in \mathcal{A}) = \underline{p_a}$. Consequently, $\mathbb{P}(Y \in \mathcal{A}) = \Phi_{r,d}\left(\Phi_{r,d}^{-1}(\underline{p_a}) - \sqrt{\delta^T \Sigma^{-1} \delta}\right)$. To ensure that $Y$ is classified as class $A$, we need

$$\Phi_{r,d}\left(\Phi_{r,d}^{-1}(\underline{p_a}) - \sqrt{\delta^T \Sigma^{-1} \delta}\right) \geq 1/2 \tag{A.24}$$

which can be satisfied if and only if $\sqrt{\delta^T \Sigma^{-1} \delta} \leq \Phi_{r,d}^{-1}(\underline{p_a}) - q_{50}$. $\qquad \square$

## A.5 Algorithm for Section 5.1

**Activation Bounds:** The dual objective function provides a bound on any linear function $c^T \hat{z}_k$. Therefore, we can compute the dual objective for $c = -I$ and $c = I$ to obtain lower and upper bounds. For $c = I$, value of $\nu_i$ for all activations simultaneously is given by

$$\hat{\nu}_i = W_i^T D_{i+1} W_{i+1}^T \ldots D_n W_n^T \text{ and } \nu_i = D_i \hat{\nu}_i, \text{ where } (D_i)_{jj} = \begin{cases} 0 & j \in \mathcal{I}_i^- \\ 1 & j \in \mathcal{I}_i^+ \\ \frac{u_{i,j}}{u_{i,j} - l_{i,j}} & j \in \mathcal{I}_i \end{cases} \tag{A.25}$$

Similar to [4], bounds for $\nu_i$ and $\hat{\nu}_i$ can be computed for each layer by cumulatively generating bounds for $\hat{z}_2$, then $\hat{z}_3$ and so on. By initializing $\hat{\nu}_1 := W_1^T$, $\zeta_1 := b_1^T$, first bounds are $l_2 := x^T W_1^T + b_1^T - \epsilon \|\Omega^{-1} W_1^T\|_q$ and $u_2 := x^T W_1^T + b_1^T + \epsilon \|\Omega^{-1} W_1^T\|_q$, where the norms are taken over the columns. Calculation of the bounds for each layer is given below in Algorithm 1.

---

**Algorithm 1** Activation Bound Calculation

---

**Input:** Network parameters $\{W_i, b_i\}$, input data $x$, input constraint matrix $\Omega$ and ball size $\epsilon$, norm type $q$.
Initialize $\hat{\nu}_1 := W_1^T$, $\zeta_1 := b_1^T$
$l_2 = x^T W_1^T + b_1^T - \epsilon \|\Omega^{-1} W_1^T\|_q$
$u_2 = x^T W_1^T + b_1^T + \epsilon \|\Omega^{-1} W_1^T\|_q$
$\nu_{2,\mathcal{I}_2} := (D_2)_{\mathcal{I}_2} W_2^T$
$\zeta_2 = b_2^T$
**for** $i = 2$ **to** $k-1$ **do**
$\qquad l_{i+1} = x^T \hat{\nu}_1 + \sum\limits_{j=1}^{i} \zeta_j - \epsilon \|\Omega^{-1} \hat{\nu}_1\|_q + \sum\limits_{i=2, i' \in \mathcal{I}_i}^{i} l_{j,i'}[-\nu_{j,i'}]_+$
$\qquad u_{i+1} = x^T \hat{\nu}_1 + \sum\limits_{j=1}^{i} \zeta_j + \epsilon \|\Omega^{-1} \hat{\nu}_1\|_q - \sum\limits_{i=2, i' \in \mathcal{I}_i}^{i} l_{j,i'}[\nu_{j,i'}]_+$
$\qquad \nu_{j,\mathcal{I}_j} = \nu_{j,\mathcal{I}_j}(D_i)_{\mathcal{I}_i} W_i^T$
$\qquad \zeta_j = \zeta_j D_i W_i^T$
$\qquad \hat{\nu}_1 = \hat{\nu}_1 (D_i)_{\mathcal{I}_i} W_i^T$
**end for**
**Output:** $\{l_i, u_i\}_{i=2}^{k}$

---

## A.6 Experiments for Section 3.1

In this section, we present a detailed explanation about the winner attacks [5] of malware competition [6], and show detailed evasion success of these attacks in Table A.1.

**GREEDY ATTACK:** Bytes in a range 256 are added iteratively to the malware binaries to make sure the prediction score for a known model lowers and none of the packing, functionality, or anti-tampering checks are affected. Byte addition is stopped when the prediction score gets lower than a threshold value or the file size exceeds 5MB. We generate 1000 adversarial examples from the malicious binaries of EMBER test set for each target model, such as standard trained neural network, adversarially trained model with $\ell_2$-PGD for $\epsilon = 5$ and LGBM model which were provided as benchmark together with EMBER dataset [2]; and we call these adversarial example sets GNN, GAdv and GLGBM, respectively.

**CONSTANT PADDING ATTACK:** A new section is created in the binary file and filled with a constant value of size 10000. This attack is applied to 2000 binaries from EMBER malicious test set for constants "169" and "0", and we call these adversarial example sets C1 Pad. and C2 Pad., respectively.

**STRING PADDING ATTACK:** Strings of size 10000 from a benign file, such as Microsoft's End User License Agreement (EULA), are added to a new section created in the malware binary. We generate 2000 adversarial examples, which we call set Str. Pad., by string padding EMBER malicious test set.

Table A.1: **Malware Use-case:** Average number of successful evasions on standard training, uniform and non-uniform $\ell_2$-PGD adversarial trainings by the adversarial example sets out of 1000 samples for approximately equal $\|\delta\|_2$. Defense success rates shown in Table A.2 and Figure 1a are calculated by averaging the success rate over these individual attacks results.

| Model | $\|\delta\|_2$ | GNN | GLGBM | GAdv | C1 Pad. | C2 Pad. | Str. Pad. |
|---|---|---|---|---|---|---|---|
| Std. Training | - | 832 | 217 | 337 | 168 | 35 | 123 |
| Uniform-$\delta$ | 0.1 | 472.6 | 105 | 249.6 | 66.3 | 37.6 | 114.9 |
| NU-$\delta$-Mask | 0.1 | 408.5 | 89.2 | 241.7 | 46.2 | 35.2 | 74.5 |
| NU-$\delta$-SHAP | 0.1 | 392.8 | 86.8 | 206.8 | 64.9 | 39 | 104.7 |
| NU-$\delta$-Pearson | 0.1 | 417.6 | 92.8 | 221.4 | 45.1 | 38.2 | 74.5 |
| NU-$\delta$-MD | 0.1 | 413 | 101 | 216 | 56 | 38.7 | 81.8 |
| NU-$\delta$-MDtarget | 0.1 | 391.6 | 84.2 | 234.6 | 52.2 | 38.7 | 79 |
| Uniform-$\delta$ | 1 | 447.2 | 111.8 | 273.8 | 50.1 | 38.5 | 83.4 |
| NU-$\delta$-Mask | 1 | 299.4 | 88.2 | 223.4 | 58.3 | 40 | 91 |
| NU-$\delta$-SHAP | 1 | 359.7 | 82.2 | 244.5 | 53.8 | 33.6 | 81.7 |
| NU-$\delta$-Pearson | 1 | 304 | 96.2 | 265 | 60.5 | 38.8 | 99.2 |
| NU-$\delta$-MD | 1 | 373.2 | 89.5 | 244 | 54.7 | 37 | 81.8 |
| NU-$\delta$-MDtarget | 1 | 360.8 | 103.4 | 246.6 | 45.1 | 36.7 | 72.4 |
| Uniform-$\delta$ | 6.7 | 231.5 | 129 | 333 | 37.7 | 38 | 58.7 |
| NU-$\delta$-Mask | 6.7 | 104.4 | 68.4 | 153.4 | 43.4 | 47.3 | 70.8 |
| NU-$\delta$-SHAP | 6.7 | 170 | 113 | 302.5 | 32.7 | 41.2 | 39.7 |
| NU-$\delta$-Pearson | 6.7 | 213 | 78 | 304 | 38 | 38 | 48.6 |
| NU-$\delta$-MD | 6.7 | 234 | 91 | 314 | 27.7 | 31 | 34.2 |
| NU-$\delta$-MDtarget | 6.7 | 196 | 61 | 301 | 37.5 | 33.5 | 36.5 |
| Uniform-$\delta$ | 11 | 177 | 77.6 | 278 | 37.8 | 41.3 | 44.6 |
| NU-$\delta$-Mask | 11 | 94.4 | 45.2 | 160.8 | 30.8 | 43.7 | 50.5 |
| NU-$\delta$-SHAP | 11 | 178 | 62 | 296 | 32 | 40 | 35 |
| NU-$\delta$-Pearson | 11 | 142 | 75.7 | 273 | 31.6 | 40.7 | 42.8 |
| NU-$\delta$-MD | 11 | 195 | 46 | 247 | 40 | 32.5 | 43 |
| NU-$\delta$-MDtarget | 11 | 122.7 | 44 | 251.7 | 34 | 41 | 48 |
| Uniform-$\delta$ | 18 | 152.2 | 57.3 | 234 | 42.7 | 51.6 | 52.3 |
| NU-$\delta$-Mask | 18 | 44.5 | 20.2 | 116.5 | 27.1 | 48.2 | 47.2 |
| NU-$\delta$-SHAP | 18 | 159.4 | 48.6 | 207.2 | 53.1 | 59.6 | 61.3 |
| NU-$\delta$-Pearson | 18 | 154.2 | 49 | 220.4 | 42.6 | 61.7 | 47.2 |
| NU-$\delta$-MD | 18 | 144.2 | 49.2 | 204.6 | 53 | 54 | 63.8 |
| NU-$\delta$-MDtarget | 18 | 132.4 | 52.4 | 215 | 50.6 | 53.4 | 53.2 |
| Uniform-$\delta$ | 25 | 233.2 | 58 | 228 | 59.3 | 51 | 68.4 |
| NU-$\delta$-Mask | 25 | 25 | 14.8 | 108 | 21.8 | 48.9 | 34.8 |
| NU-$\delta$-SHAP | 25 | 193.8 | 53.4 | 226.2 | 44.7 | 56.9 | 58.7 |
| NU-$\delta$-Pearson | 25 | 158.2 | 59.5 | 191.2 | 67.6 | 65.6 | 75.8 |
| NU-$\delta$-MD | 25 | 199.7 | 54 | 248.5 | 58.6 | 59.5 | 59.7 |
| NU-$\delta$-MDtarget | 25 | 210 | 55 | 225.6 | 57.7 | 56.4 | 60.5 |

Table A.1 shows the average number of adversarial examples out of 1000 which successfully evade the corresponding models. While *NU-δ-Mask* and *NU-δ-MDtarget* have better performance against Greedy attacks for most of the time, i.e., sets GNN, GLGBM and GAdv, *NU-δ-Pearson*, *NU-δ-SHAP* and *NU-δ-MD* have better accuracy against padding attacks, i.e, sets C1 Pad., C2 Pad. and Str. Pad.

Table A.2: **Malware Use-case:** Clean accuracy (Ac.) and defense success rate (S.R.) of standard training, uniform and non-uniform $\ell_2$-PGD adversarial trainings with EMBER dataset for approximately equal $\|\delta\|_2$. Non-uniform perturbation defense approaches outperform the uniform perturbation for all cases against adversarial attacks.

| Model | $\|\delta\|_2$ | Clean Ac., % | Defense S.R., % |
|---|---|---|---|
| Std. Training | - | 96.6 | 73 |
| Uniform-$\delta$ | 0.1 | 96.2 | $82.7 \pm 0.88$ |
| NU-$\delta$-Mask | 0.1 | 96.2 | $85.3 \pm 0.25$ |
| NU-$\delta$-SHAP | 0.1 | 96.2 | $85.2 \pm 0.39$ |
| NU-$\delta$-Pearson | 0.1 | 96.1 | $85.3 \pm 0.94$ |
| NU-$\delta$-MD | 0.1 | 96.3 | $85 \pm 0.99$ |
| **NU-$\delta$-MDtarget** | 0.1 | 96.2 | $\mathbf{85.4 \pm 0.80}$ |
| Uniform-$\delta$ | 1 | 96.1 | $83.3 \pm 0.41$ |
| **NU-$\delta$-Mask** | 1 | 96.1 | $\mathbf{86.7 \pm 0.68}$ |
| NU-$\delta$-SHAP | 1 | 96.3 | $85.5 \pm 0.61$ |
| NU-$\delta$-Pearson | 1 | 96.2 | $85.7 \pm 0.45$ |
| NU-$\delta$-MD | 1 | 96.3 | $85.4 \pm 0.67$ |
| NU-$\delta$-MDtarget | 1 | 96.3 | $85.9 \pm 0.19$ |
| Uniform-$\delta$ | 6.7 | 95.8 | $86.3 \pm 0.15$ |
| **NU-$\delta$-Mask** | 6.7 | 95.7 | $\mathbf{92 \pm 0.07}$ |
| NU-$\delta$-SHAP | 6.7 | 95.8 | $88.3 \pm 0.33$ |
| NU-$\delta$-Pearson | 6.7 | 95.9 | $88.2 \pm 0.30$ |
| NU-$\delta$-MD | 6.7 | 96 | $87.7 \pm 0.32$ |
| NU-$\delta$-MDtarget | 6.7 | 95.8 | $89 \pm 0.18$ |
| Uniform-$\delta$ | 11 | 95.6 | $89.3 \pm 0.54$ |
| **NU-$\delta$-Mask** | 11 | 95.8 | $\mathbf{92.9 \pm 0.57}$ |
| NU-$\delta$-SHAP | 11 | 96 | $90.3 \pm 0.36$ |
| NU-$\delta$-Pearson | 11 | 95.8 | $90 \pm 0.36$ |
| NU-$\delta$-MD | 11 | 95.9 | $89.9 \pm 0.25$ |
| NU-$\delta$-MDtarget | 11 | 95.7 | $90.9 \pm 0.29$ |
| Uniform-$\delta$ | 18 | 95.5 | $90.17 \pm 0.71$ |
| **NU-$\delta$-Mask** | 18 | 95.8 | $\mathbf{94.8 \pm 0.51}$ |
| NU-$\delta$-SHAP | 18 | 95.3 | $90.45 \pm 0.30$ |
| NU-$\delta$-Pearson | 18 | 95.3 | $90.46 \pm 0.25$ |
| NU-$\delta$-MD | 18 | 95.4 | $90.54 \pm 0.46$ |
| NU-$\delta$-MDtarget | 18 | 95.4 | $90.7 \pm 0.51$ |
| Uniform-$\delta$ | 25 | 95.6 | $88.4 \pm 0.39$ |
| **NU-$\delta$-Mask** | 25 | 95.7 | $\mathbf{95.8 \pm 0.21}$ |
| NU-$\delta$-SHAP | 25 | 95.5 | $89.5 \pm 0.27$ |
| NU-$\delta$-Pearson | 25 | 94.9 | $89.7 \pm 0.40$ |
| NU-$\delta$-MD | 25 | 95.2 | $88.6 \pm 0.26$ |
| NU-$\delta$-MDtarget | 25 | 95.2 | $89 \pm 0.57$ |

Table A.3: **Credit Risk Use-case:** Clean accuracy (Ac.) and defense success rate (S.R.) of standard training, uniform and non-uniform $\ell_2$-PGD adversarial trainings with German Credit dataset for approximately equal $\|\delta\|_2$. Non-uniform perturbation defense approaches outperform the uniform perturbation for all cases against adversarial attacks.

| Model | $\|\delta\|_2$ | Clean Ac., % | Defense S.R., % |
|---|---|---|---|
| Std. Training | - | 69.7 | 60 |
| Uniform-$\delta$ | 0.01 | 69 | $61.3 \pm 0.40$ |
| NU-$\delta$-SHAP | 0.01 | 68.3 | $61.3 \pm 0.35$ |
| **NU-$\delta$-Pearson** | 0.01 | 68.3 | $\mathbf{61.9 \pm 0.37}$ |
| NU-$\delta$-MD | 0.01 | 69.6 | $61.6 \pm 0.32$ |
| **NU-$\delta$-MDtarget** | 0.01 | 69.7 | $\mathbf{61.9 \pm 0.30}$ |
| Uniform-$\delta$ | 0.1 | 67.7 | $63.4 \pm 0.31$ |
| **NU-$\delta$-SHAP** | 0.1 | 67.1 | $\mathbf{64.5 \pm 0.20}$ |
| NU-$\delta$-Pearson | 0.1 | 66.8 | $64.3 \pm 0.56$ |
| NU-$\delta$-MD | 0.1 | 66.7 | $64.2 \pm 0.32$ |
| **NU-$\delta$-MDtarget** | 0.1 | 66.7 | $\mathbf{64.5 \pm 0.41}$ |
| Uniform-$\delta$ | 0.3 | 66.7 | $66.4 \pm 0.22$ |
| NU-$\delta$-SHAP | 0.3 | 65.8 | $67.6 \pm 0.30$ |
| NU-$\delta$-Pearson | 0.3 | 66 | $68 \pm 0.21$ |
| NU-$\delta$-MD | 0.3 | 66.5 | $67.1 \pm 0.64$ |
| **NU-$\delta$-MDtarget** | 0.3 | 66.3 | $\mathbf{69 \pm 0.32}$ |
| Uniform-$\delta$ | 0.5 | 66.2 | $68 \pm 0.32$ |
| NU-$\delta$-SHAP | 0.5 | 66.5 | $69.7 \pm 0.37$ |
| NU-$\delta$-Pearson | 0.5 | 65.9 | $69.4 \pm 0.27$ |
| NU-$\delta$-MD | 0.5 | 66.3 | $69.2 \pm 0.35$ |
| **NU-$\delta$-MDtarget** | 0.5 | 66 | $\mathbf{69.8 \pm 0.13}$ |
| Uniform-$\delta$ | 0.7 | 66.1 | $69.6 \pm 0.20$ |
| **NU-$\delta$-SHAP** | 0.7 | 65.8 | $\mathbf{71.1 \pm 0.57}$ |
| NU-$\delta$-Pearson | 0.7 | 65.6 | $71 \pm 0.37$ |
| NU-$\delta$-MD | 0.7 | 66.4 | $70.5 \pm 0.30$ |
| NU-$\delta$-MDtarget | 0.7 | 65.6 | $70.3 \pm 0.30$ |
| Uniform-$\delta$ | 1 | 65.3 | $70.6 \pm 0.44$ |
| **NU-$\delta$-SHAP** | 1 | 64.5 | $\mathbf{71.3 \pm 0.32}$ |
| **NU-$\delta$-Pearson** | 1 | 64.3 | $\mathbf{71.3 \pm 0.32}$ |
| NU-$\delta$-MD | 1 | 64.9 | $71 \pm 0.37$ |
| NU-$\delta$-MDtarget | 1 | 65 | $71 \pm 0.21$ |

Table A.4: **Spam Detection Use-case:** Clean accuracy and defense success rate of standard training, uniform and non-uniform $\ell_2$-PGD adversarial trainings with Twitter Spam dataset for approximately equal $\|\delta\|_2$.

| Model | $\|\delta\|_2$ | Clean Ac., % | Defense S.R., % |
|---|---|---|---|
| Std. Training | - | 94.6 | 17.5 |
| Uniform-$\delta$ | 0.1 | 91.1 | $34.4 \pm 0.16$ |
| NU-$\delta$-SHAP | 0.1 | 93.9 | $35.3 \pm 0.32$ |
| NU-$\delta$-Pearson | 0.1 | 94 | $36 \pm 0.50.$ |
| NU-$\delta$-MD | 0.1 | 93.9 | $36.7 \pm 0.48$ |
| **NU-$\delta$-MDtarget** | 0.1 | 93.9 | $\mathbf{38.3 \pm 0.50}$ |
| Uniform-$\delta$ | 0.3 | 92.6 | $58.3 \pm 0.66$ |
| NU-$\delta$-SHAP | 0.3 | 91.9 | $66.5 \pm 0.86$ |
| NU-$\delta$-Pearson | 0.3 | 91.8 | $65 \pm 0.21$ |
| **NU-$\delta$-MD** | 0.3 | 91.9 | $\mathbf{69.4 \pm 0.25}$ |
| NU-$\delta$-MDtarget | 0.3 | 92 | $67.9 \pm 0.25$ |
| Uniform-$\delta$ | 0.5 | 91.3 | $82.8 \pm 0.46$ |
| NU-$\delta$-SHAP | 0.5 | 90.9 | $86.1 \pm 0.14$ |
| **NU-$\delta$-Pearson** | 0.5 | 91.2 | $\mathbf{87.3 \pm 0.20}$ |
| NU-$\delta$-MD | 0.5 | 91.1 | $85.3 \pm 0.30$ |
| NU-$\delta$-MDtarget | 0.5 | 91.2 | $86.8 \pm 0.28$ |
| Uniform-$\delta$ | 0.7 | 91.1 | $89.6 \pm 0.48$ |
| NU-$\delta$-SHAP | 0.7 | 90.5 | $90.5 \pm 0.19$ |
| **NU-$\delta$-Pearson** | 0.7 | 90.6 | $\mathbf{90.7 \pm 0.11}$ |
| NU-$\delta$-MD | 0.7 | 90.5 | $89.8 \pm 0.35$ |
| NU-$\delta$-MDtarget | 0.7 | 90.5 | $89.1 \pm 0.18$ |
| Uniform-$\delta$ | 1 | 90.5 | $87.3 \pm 0.62$ |
| NU-$\delta$-SHAP | 1 | 89.8 | $91.4 \pm 0.62$ |
| NU-$\delta$-Pearson | 1 | 89.9 | $92 \pm 0.53$ |
| **NU-$\delta$-MD** | 1 | 89.7 | $\mathbf{93.3 \pm 0.30}$ |
| NU-$\delta$-MDtarget | 1 | 89.8 | $92.5 \pm 0.64$ |

## A.7 Experiments for Section 3.4

We further investigate the performance of our non-uniform approach against uniformly norm-bounded attacks for generalizability as in Section 3.4. We use the same setting as in spam detection use-case, and craft AEs using standard PGD attack, i.e., the attack in *Uniform-$\delta$*, for $\epsilon = \{0.1, 0.3, 0.5, 0.7\}$. For a fair comparison between the uniform and non-uniform approaches, we set approximately equal $\|\delta\|_2$ for both models in the average sense. In this section, we also consider a non-uniform robust model which enforces the AT constraint first on $\|\Omega\delta\|_2$ and then $\|\delta\|_2$. That is, the non-uniform attack is always a valid uniform attack in the strict sense. We call this defense *Combo* due to using the combination of both projections in (3) and (5).

Table A.5 shows the defense success rates of *Uniform-$\delta$*, *NU-$\delta$-MDt* and *MDt-Combo*, which denotes the *Combo* approach for $\Omega$ selected as the Mahalanobis matrix for the benign samples, against PGD attacks for Spam Detection Use-case. We observe that our non-uniform approach outperforms the uniform approach for all cases, hence it is also effective against the uniformly norm-bounded attacks which makes it generalizable. Furthermore, Table A.5 shows that *MDt-Combo* performs in between *Uniform-$\delta$* and *NU-$\delta$-MDt*. This is due to the fact that the strict constraint on $\|\delta\|_2$ reduces the effect of non-uniform projection.

Table A.5: Defense success rates of *Uniform-δ*, *NU-δ-MDt* and *MDt-Combo* against PGD attacks for Spam Detection Use-case. Both non-uniform defenses outperform the uniform approach while *NU-δ-MDt* also outperforms *MDt-Combo* for all cases.

| Defenses | $||\delta||_2$ | Attack $\epsilon$=0.1 | Attack $\epsilon$=0.3 | Attack $\epsilon$=0.5 | Attack $\epsilon$=0.7 |
|---|---|---|---|---|---|
| Uniform-δ | | $90.6 \pm 0.18$ | $83.9 \pm 0.34$ | $16.6 \pm 0.47$ | $14.8 \pm 0.65$ |
| NU-δ-MDt | 0.1 | $\mathbf{92.8 \pm 0.21}$ | $\mathbf{88.2 \pm 0.41}$ | $\mathbf{34.95 \pm 0.5}$ | $\mathbf{21.4 \pm 0.22}$ |
| MDt-Combo | | $91.6 \pm 0.24$ | $86.3 \pm 0.28$ | $24.4 \pm 0.38$ | $19.2 \pm 0.32$ |
| Uniform-δ | | $92.6 \pm 0.14$ | $89.05 \pm 0.24$ | $30.5 \pm 0.42$ | $20.85 \pm 0.58$ |
| NU-δ-MDt | 0.3 | $\mathbf{93.2 \pm 0.10}$ | $\mathbf{90.95 \pm 0.22}$ | $\mathbf{61.75 \pm 0.51}$ | $\mathbf{46.3 \pm 0.41}$ |
| MDt-Combo | | $93 \pm 0.11$ | $89.24 \pm 0.19$ | $47.88 \pm 0.72$ | $31.5 \pm 0.39$ |
| Uniform-δ | | $92.9 \pm 0.08$ | $91.4 \pm 0.05$ | $88 \pm 0.22$ | $86.5 \pm 0.25$ |
| NU-δ-MDt | 0.5 | $\mathbf{93.3 \pm 0.05}$ | $\mathbf{91.45 \pm 0.06}$ | $\mathbf{89.45 \pm 0.17}$ | $\mathbf{87.7 \pm 0.12}$ |
| MDt-Combo | | $93.1 \pm 0.10$ | $91.4 \pm 0.11$ | $89.30 \pm 0.14$ | $87.2 \pm 0.18$ |
| Uniform-δ | | $93.2 \pm 0.14$ | $91.9 \pm 0.25$ | $90.18 \pm 0.20$ | $88.38 \pm 0.22$ |
| NU-δ-MDt | 1 | $\mathbf{94.5 \pm 0.11}$ | $\mathbf{94.17 \pm 0.42}$ | $\mathbf{93.33 \pm 0.15}$ | $\mathbf{93.14 \pm 0.31}$ |
| MDt-Combo | | $94.39 \pm 0.10$ | $92.42 \pm 0.28$ | $91.44 \pm 0.19$ | $91.33 \pm 0.26$ |
| Uniform-δ | | $93.22 \pm 0.16$ | $92.01 \pm 0.27$ | $90.61 \pm 0.24$ | $89.78 \pm 0.15$ |
| NU-δ-MDt | 1.5 | $\mathbf{94.45 \pm 0.12}$ | $\mathbf{94.38 \pm 0.31}$ | $\mathbf{93.76 \pm 0.16}$ | $\mathbf{93.48 \pm 0.11}$ |
| MDt-Combo | | $94.27 \pm 0.11$ | $93.1 \pm 0.20$ | $92.3 \pm 0.27$ | $91.8 \pm 0.17$ |