# OpenReview forum: "Adversarial Robustness with Non-uniform Perturbations"
_NeurIPS.cc/2021/Conference — NeurIPS 2021 Poster_

### Official Review · Reviewer_Dp1A · 2021-06-28

**Rating:** 7
**Confidence:** 5

**Summary:**

The paper tackles the problem of adversarial attacks against DNNs. The paper revolves around “non-uniform” perturbations: in traditional image-based attacks, it is possible to craft an adversarial sample without considering “inter-feature dependencies”. However, in other application domains where ML can be used, such freedom is not possible, which prevents the application of popular countermeasures, such as Adversarial Training (AT). The paper thus investigates the application of AT in these constrained perturbations scenarios, and compares the effectiveness of AT in “unconstrained” perturbations against the (proposed) “constrained” perturbations.

Overall, the presentation of the paper is excellent.
The quality of the English text is excellent.
Figures and Tables are very good.
The topic addressed by the manuscript is intriguing because it takes into account real-world constraints.
The references are appropriate.
The contribution is small, but significant.

STRENGTHS:
+ Very well written
+ Consideration of real world constraints
+ Evaluation on multiple datasets of distinct domains (Malware, Bank Fraud, Spam)
+ Technically sound
+ Good organization

WEAKNESSES
- The considered attacks are performed in the feature space (AFTER REBUTTAL: this is not a problem anymore)


**Limitations And Societal Impact:**

The authors should clearly say that the considered attacks are not truly in the problem space and that they may not be what a true attacker would do in reality.

This is UNLESS the authors decide to thoroughly explain and motivate that I what I wrote is incorrect.


AFTER REBUTTAL: the authors clarified that the attacks are in the problem space.

**Main Review:**

I very much liked reading this paper. Unlike “normal” papers on adversarial attacks, what sets it apart is assuming a “more realistic” perspective where the perturbations are subject to realistic constraints (feature dependencies) and the evaluation is conducted on datasets that are not image-related. The combination of these two factors is a very good contribution to the state of the art, because there are many domains where ML can be applied and there is an over-abundance of papers that focus exclusively on computer vision tasks. I also appreciated the overall organization of the paper. The first section is concise and on point, and properly introduces the reader to the subject. In the second section the “proposed method” is explained and the description is technically sound. Then, the third and fourth sections are devoted to the experimental analysis and results, and the evaluation also considers Certified Defenses.

I have only one problem with this paper, and it’s related to the fact that the considered attacks are not performed in the problem space (contrarily to what the authors claim). All experiments rely on datasets that contain “pre-collected” data. Hence, to craft the considered adversarial samples, the authors use such data as base. However, we cannot be certain that such perturbations can truly reflect the actions of a true attacker that performs an adversarial attack in the real world. In other words, I am referring to the fact that the AE are created in the feature space.
It would have been very interesting if – among the current experiments – there was an additional use case where the authors use problem-space AE to clearly highlight the key differences between uniform and non-uniform perturbations. I think that a similar addition would dramatically improve the contribution and overall quality of the paper. However, I do not think that the Ember dataset is sufficient to do so (more on this part below). And the other 2 use cases also are not problem space attacks, from what I can understand by the (very limited) information provided.

In summary, I consider the paper to be in a good state: the presentation and contribution are good enough to recommend a “weak/regular” acceptance. The addition of a (even as a proof-of-concept) use case with a true problem-space adversarial examples would immediately make my recommendation become a “strong accept”.


Explanation on EMBER Use Case: In Section 3.1, the authors clearly state that they “We evaluate the robustness of our model against evasion attacks which are crafted in problem-space, i.e., on PE files”. Further on, in lines 209-211, the authors state the following: “We incorporate the most successful attacks [33] from the machine learning static evasion competition of [34]. Since the EMBER dataset only contains the extracted features of a file, a subset of malware binaries used for AE generation are obtained from VirusTotal [35] using the SHA-256 hash as identifier.”
I believe this experimental methodology to be one of the following: flawed, unclear, or not representative of a problem space attack.
In a problem space (malware) attack, the adversary modifies the malware binary. However, I do not see anything in the above description that leads me to believe that this is what the authors did. What the authors should do is take an existing piece of malware executable, modify it, obtain its feature-like representation (using the same structure as in Ember), and then insert it into the Ember dataset. However, as mentioned by the very same authors (in line 199), Ember is a “feature-based dataset”. Hence it is not possible to perform “problem space” attacks using Ember as source. The best that can be done using Ember is operating at the “raw data” space – which is closer to reality than the higher level “feature space”, but definitely not in the “problem space”.
All of the above should be clarified in the paper.
Regardless, all of these details should be added into the main body of the paper. What kind of manipulations did the authors introduce? Are these manipulations “realistically” feasible? Do they represent a “likely” behaviour of a realistic attacker who is trying to evade detection by minimally changing the malware executable? The authors do provide some external references, but they should further describe what is being done in their experiments: by looking at those references, the only (relevant) information I get is that “the attacks must satisfy the physical constraints of PE malware”, but there is no specification of whether the considered attack strategy is either viable or realistically sensible.

Some minor remarks:

•	The paragraph from lines 129 to 134 could be better structured. It has lengthy notation that breaks through lines, hence I suggest a revisioning to improve the readability

•	Figure 1 is very hard to read (I suggest using different colors for all lines)

•	I would like the authors to consider the related work in [A] (see below) and explain how their paper can be relevant to the scenario envisioned in [A].


EXTERNAL REFERENCES

[A]: Erba, Alessandro, et al. "Constrained Concealment Attacks against Reconstruction-based Anomaly Detectors in Industrial Control Systems." Annual Computer Security Applications Conference. 2020.

_________________________--

AFTER REBUTTAL: the authors clarified that the attacks are in the problem space. I will raise my score from 6 to 7.

**Time Spent Reviewing:**

4

---

> ### Author Response · Authors · 2021-08-09
> **Response to the Reviewer Dp1A**
>
> We would like to thank the reviewer for careful reading of the paper and encouraging comments.
>
> - Problem space attacks and explanation of EMBER
>
> Although our defense is an adversarial training (AT) approach based on feature space perturbations during training, we test our defense against various “realistic attacks” such as problem space based malware evasion attacks described in Appendix A.6, and feature space attacks for credit risk and spam detection use-cases. It is worth remembering that sophisticated feature space attacks can still be realistic if the features are the real-world object itself as in the case of credit-risk detection.
>
> We tested our defense with various real-world attacks which were known to be effective in the literature, i.e., DEFCON competition winner malware attacks in Appendix A.6, LowProFool [17], and etc. Our feature space defense is more advantageous when there is limited domain knowledge or access to a large representative corpus of binary files is not possible as we explained in 98-103. However, we **indeed** crafted real AEs in the problem space in malware use-case **as the reviewer suggested**. For this, we used real malware binaries from the EMBER dataset. EMBER is provided as a featurized version of real malware binaries, however, the real PEs can be downloaded in batches from VirusTotal using the provided SHA-256 identifier by premium users. We modified real binaries by padding bytes to the overlay of the file, and injected sections containing digits and benign strings from Microsoft EULA. These attacks are feasible and do not break the functionality of the PEs since the padded bytes and the new sections are not executed. Finally, we converted these files into features as the way the entire EMBER dataset is converted. This transformation function for featurization is provided on the EMBER dataset’s github page. Our malware use-case attacks are exact problem space attacks, but our neural network is fed with a featurized version of these malicious binaries in test time. We provide these featurized AEs in our supplementary material.
>
> - Minor remarks
>
> We will update the lines 129-134 and Figure 1 accordingly. We also thank the reviewer for the reference [A]. We have not considered attacks to anomaly detection systems but it could be a new application area of our approach.

---

> > ### Comment · Reviewer_Dp1A · 2021-08-11
> > **Good response**
> >
> > I thank the authors for their response which clarified my (only) doubt.
> >
> > I ***STRONGLY*** enforce to better remark (everywhere) the fact that the attacks are performed in the problem space. There is a lack of problem-space attacks in adversarial ML literature, and - despite being only experimental - this is a significant contribution.
> >
> > Moreover, the explanation provided in this comment should be inserted somewhere in the paper (even the appendix would suffice, but in the main text it should be clearly stated that the attack is in the problem space, and that the detailed methodology is presented in the appendix).

---

> > > ### Author Response · Authors · 2021-08-12
> > > **Response to the Reviewer's Comment**
> > >
> > > We would like to thank the reviewer for encouraging comments. Our proposed defense targets realistic attacks, such as problem space attacks in the malware domain. We will clarify this by including our response in the paper as the reviewer suggests.

---

> > > > ### Comment · Reviewer_Dp1A · 2021-08-25
> > > > **Baseline performance change**
> > > >
> > > > After re-reading the paper, I have an additional question that I would love to see an answer to, and which would increase the "practicality" of the paper.
> > > >
> > > > Does the application of (any) AT result in a degradation of the performance on "clean" samples (i.e., non-adversarially manipulated)? I cannot find any such result in the paper. Many papers pointed out that countermeasures to Adversarial Attack can degrade baseline performance (e.g. [A]). This could be another interesting insight that can be shown by the paper.
> > > >
> > > > Hence, I ask you: am I right, or was this "degradation" (or improvement) mentioned in the paper somewhere? If I'm right and it is indeed missing, then I invite the author to assess such circumstances. If they are already present, then they should be given more emphasis.
> > > >
> > > > [A] "Deep Reinforcement Adversarial Learning against Botnet Evasion Attacks" IEEE Transactions on Network and Service Management, 2020

---

> > > > > ### Author Response · Authors · 2021-08-25
> > > > > **Response to the Reviewer's Comment**
> > > > >
> > > > > We would like to thank the reviewer for noticing the need for emphasizing our experimental results for clean accuracy. We mention the accuracy of an adversarially trained model on unattacked samples as “clean accuracy” in Sections 3.1, 3.2 and 3.3, and we refer to the results in Tables A.2 A.3 and A.4. We also provided the clean accuracy of only benign samples in the table posted as a response to the Reviewer Y8xB. Overall the results show that any AT causes some level of clean accuracy loss, but this degradation is not large compared to the gain from AT against attacks. We will clarify these in the main body of the paper.

---

> > > > > > ### Comment · Reviewer_Dp1A · 2021-08-25
> > > > > > **Thanks**
> > > > > >
> > > > > > Thanks for the clarification; the Appendix does indeed contain such information. I believe the authors should emphasize its significance.
> > > > > >
> > > > > > I acknowledge that AT (with or without uniform perturbations) can be a good way to protect against AA; however, we must always consider that (i) adversarial examples will represent a minority of examples analyzed by any ML algorithm in reality; and that (ii) AT (in any of its variant) can only protect against those adversarial attacks whose samples "conform" to a specific attack pattern. It is thus crucial to put a great emphasis on the effectiveness of AT in the absence of adversarial attacks.
> > > > > >
> > > > > > I think that including such remarks in the main paper (even if the results are shown in the appendix) could better improve the contribution of the paper to the State of the Art, as it would allow to investigate the effectiveness of AT under different circumstances. It is still empyrical and dataset-based, but these results are still useful.

---

> > > > > > > ### Author Response · Authors · 2021-08-25
> > > > > > > **Response to the Reviewer's Comment**
> > > > > > >
> > > > > > > We strongly agree with the reviewer that clean accuracy needs more emphasis in the main body of the paper. We will update the **Adversarial Training** part in Section 3 by adding remarks on clean accuracy, and also emphasize its empirical results in Sections 3.1, 3.2, and 3.3 accordingly. We would like to thank the reviewer for this suggestion.

---

### Official Review · Reviewer_tqQw · 2021-06-30

**Rating:** 7
**Confidence:** 4

**Summary:**

The standard adversarial attack is done against some p-norm bounded radius, which implicitly assumes each dimension of the attack space is equally valid. This is not always true, and this paper proposes adversarial attacks (and training) with quadratic relationship constraints on the usage of each feature in the attack generation process via a covariance matrix applied to the perturbation vector. Improved adversarial training is shown across several datasets and prior results on certifiable robustness are extended to the paper's domain.

**Ethical Concerns:**

I have no special ethical concerns with regard to this work. Overall it is improving defensive posture, and adversarial attack/defense in ML is hard to seperate.

**Limitations And Societal Impact:**

The paper could have better emphasized that this approach will not be applicable to all problem spaces, but that is implied. I'm fine with it but I could see some readers needing it stated more explicitly.

**Main Review:**

I overall like the paper, the need to consider attacks adjusted to specific features is intuitive and makes sense. The results showing how the attacks end up looking more like benign samples is compelling, and the extension of the results to certifiable bounds in two different ways is good. Overall a thorough and well done paper.

My only major concern is if the comparison between new and prior PGD with respect to a maximum perturbation ε was done in a fully apples-to-apples maner. By using ||Ωδ||_p ≤ ε , but not constraining Ω to maintain the norm of δ, I fear that things are not quite equal. I think two items could fully dissuade this:

1. Results when the ||Ωδ||_p attack is projected at every step to be within the ||δ||_p ball, so that the ||Ωδ||_p attack is always a valid ||δ||_p  attack. (It may be that Figure 1 shows this but if so it was not clearly explained)
2. Showing results of the model training under ||Ωδ||_p adversarial training and it's accuracy against ||δ||_p generated attacks.

I do have some minor points around the malware /executable sections of the document. I think the simplifications are fine, but it should be stated that the features that could be altered is larger than what is considered through other approaches such as packing, binary rewriting, and other obfuscations. The line has to be drawn somewhere, as these are not always practical attacks, but they do exist. You could use [4] to support the large degree of complexity there. [5] should also be at least discussed in the related work around feature based constraints on the model attack space.

In particular 51-56, 181-183 should be referencing [1] and [2] which independently developed monotonic approaches to malware detection under the "additive" threat model, which incorporates malware and spam specific constraints on the attack space into the defense. This relates to the classic "good word" attack for spam [3] that should also be referenced appropriately in that portion.

My Citations:
1. Incer, I., Theodorides, M., Afroz, S., & Wagner, D. (2018). Adversarially Robust Malware Detection Using Monotonic Classification. In Proceedings of the Fourth ACM International Workshop on Security and Privacy Analytics (pp. 54–63). New York, NY, USA: ACM. https://doi.org/10.1145/3180445.3180449
2. Fleshman, W., Raff, E., Sylvester, J., Forsyth, S., & McLean, M. (2019). Non-Negative Networks Against Adversarial Attacks. AAAI-2019 Workshop on Artificial Intelligence for Cyber Security. Retrieved from http://arxiv.org/abs/1806.06108
3. Lowd, D., & Meek, C. (2005). Adversarial Learning. In Proceedings of the Eleventh ACM SIGKDD International Conference on Knowledge Discovery in Data Mining (pp. 641–647). New York, NY, USA: ACM. https://doi.org/10.1145/1081870.1081950
4. Raff, E., & Nicholas, C. (2020). A Survey of Machine Learning Methods and Challenges for Windows Malware Classification. In NeurIPS 2020 Workshop: ML Retrospectives, Surveys & Meta-Analyses (ML-RSA). Retrieved from http://arxiv.org/abs/2006.09271
5. Demontis, A., Melis, M., Biggio, B., Maiorca, D., Arp, D., Rieck, K., … Roli, F. (2017). Yes, Machine Learning Can Be More Secure! A Case Study on Android Malware Detection. IEEE Transactions on Dependable and Secure Computing, 1–1. https://doi.org/10.1109/TDSC.2017.2700270

-------

I believe incorporating the results in rebuttal to my review are satisficing, and have adjusted my score appropriately.

**Time Spent Reviewing:**

3

---

> ### Author Response · Authors · 2021-08-09
> **Response to the Reviewer tqQw**
>
> We would like to thank the reviewer for the careful reading and encouraging comments.
>
> - comparison between new and prior PGD and results where $\|\|\Omega\delta\|\|_p$ attack is always a valid $\|\|\delta\|\|_p$ attack
>
> The constraints of the new PGD and the prior one are $|| \Omega \delta ||_p \leq \epsilon_1$ and $|| \delta ||_p \leq \epsilon_2$, respectively. We choose $\epsilon_1$ and  $\epsilon_2$ such that both PGDs have approximately equal $|| \delta ||_p$.
> This is also shown in Figure 1 and on Tables A.1-A.4. Hence, the $|| \Omega \delta ||_p$ attack is always a valid $|| \delta ||_p$ attack as the reviewer states in point 1. We will clarify this in the paper.
>
> - Testing against $\|\|\delta\|\|_p$ generated attacks
>
> We performed extra experiments on $\|\|\delta\|\|_p$ generated attacks. Below are the few results on the defense success rate of Uniform-$\delta$ and NU-$\delta$-MDtarget against uniform PGD attack for various perturbations. The table shows that our non-uniform approach is more robust than the uniform approach even against uniform PGD attacks.
>
> |  Defenses              | $\delta$ | Uniform Attack $\epsilon$=0.1 | Uniform Attack $\epsilon$=0.3 | Uniform Attack $\epsilon$=0.5 | Uniform Attack $\epsilon$=0.7 |
> |------------------------|--------|-----------------------------|-----------------------------|-----------------------------|-----------------------------|
> | Uniform-$\delta$         | 0.1    | 90.6                        | 83.9                        | 16.6                        | 14.8                        |
> | NU-$\delta$-MDtarget     | 0.1    | 92.8                        | 88.2                        | 34.95                       | 21.4                        |
> | MDtarget-Combo         | 0.1    | 91.6                        | 86.3                        | 24.4                        | 19.2                        |
> | Uniform-$\delta$         | 0.3    | 92.6                        | 89.05                       | 30.5                        | 20.85                       |
> | NU-$\delta$-MDtarget     | 0.3    | 93.2                        | 90.95                       | 61.75                       | 46.3                        |
> | MDtarget-Combo         | 0.3    | 93                          | 89.24                       | 47.88                       | 31.5                        |
> | Uniform-$\delta$         | 0.5    | 92.9                        | 91.4                        | 88                          | 86.5                        |
> | NU-$\delta$-MDtarget    | 0.5    | 93.3                        | 91.45                       | 89.45                       | 87.7                        |
> | MDtarget Combo | 0.5    | 93.1                        | 91.4                        | 89.30                       | 87.2                        |
>
> - Minor points
>
> We would like to thank the reviewer for the references, and we will certainly add more discussions on the feature and problem space by referring to the suggested works [4] and [5] accordingly, as well as to [1-3].

---

> > ### Comment · Reviewer_tqQw · 2021-08-10
> > **Why Does This Need a New Title?**
> >
> > Ok, I think I understand Figure 1 a bit better. It is common that many attacks do not use the full $\epsilon$ budget, I thought Figure 1's "Average $\delta$" was referring to the norm that actually occurred and not the budget. I think the approach taken then is valid, but needs to be very clearly stated - as it is unusual. I think I would still prefer, and I think less confusing, would be to project so that the $\|\|\Omega\delta\|\|_p$ attack is forced to be an always valid $\|\|\delta\|\|_p$ attack, as the current setup is only valid in expectation/on average.
> >
> > Could the authors clarify the difference between $\|\|\delta\|\|_2$ = {0.1, 0.3, 0.5} in this table compared to the $\epsilon$={0.1, 0.3, 0.5, 0.7}? Is this adversarial training $\epsilon$ vs attack-time $\epsilon$?

---

> > > ### Author Response · Authors · 2021-08-10
> > > **Response to the Reviewer's Comment**
> > >
> > > We will make sure our approach for fair comparison is clearly stated in the paper. We thank the reviewer for the suggestion, but enforcing a strict $\delta$ constraint on top of our $\Omega \delta$ constraint at each iteration would mutilate the effect of Mahalanobis transformation or important features.
> > >
> > > $\epsilon$ is the typical adversarial training parameter, not the attack time. We used $||\delta||_2$ notation for all our defenses, because it is the equal budget enforced for fair comparison in the average sense. Therefore, we represented them separately.

---

> > > > ### Comment · Reviewer_tqQw · 2021-08-12
> > > > **Add projection failure results? I still don't like having to type a new title for every message.**
> > > >
> > > > Can the authors share results on the projection approach not working? I think it's a natural question the reader may ask (though I am certainly biased since it was my question), and I think could make the manuscript stronger to show alternative methodologies attempted but did not become fruitful.

---

> > > > > ### Author Response · Authors · 2021-08-12
> > > > > **Response to the reviewer’s experimental result request**
> > > > >
> > > > > We included new results in the table of the first response and named the approach with both $\Omega\delta$ and $\delta$ constraints as MDtarget-Combo. Based on the results, we cannot say the strict constraint on $\delta$ is not working, but the effect of the NU-$\delta$-MDtarget  approach is reduced. MDtarget-Combo performance is between the uniform projection and the nonuniform projection, which is expected as a result of replacing the average constraint on the perturbations with a more strict constraint. We will add a discussion on this in the paper, and mention the experimental results as well. Again, we thank the reviewer for this suggestion.

---

### Official Review · Reviewer_8HjK · 2021-07-01

**Rating:** 6
**Confidence:** 4

**Summary:**

Most research on adversarial perturbations is focused on those within an lp-ball (termed "uniform" due to the assumption that relative perturbation costs are uniform within the ball).  This paper introduces a non-uniform model of perturbations aimed to better capture feature dependencies and correlations through a space of better, non-uniform, cost functions.  The main goal is to use the modified attack model as a part of AT, with the hypothesis that the resulting AT will be more effective against problem-space attacks (commonly also called realizable attacks in prior literature).

**Main Review:**

The paper hits an important gap in the literature.  However, it seems to miss some of the closely related work in the malware and image classification space.  Moreover, while the authors appear to make a conceptual distinction between attacks on images and domains such as malware/spam/fraud detection, the distinction seems unnecessary.  Given prior work that had already studied the problem of realizable or problem-space attacks in the context of AT, the conceptual contribution appears limited.  Furthermore, the technical contribution is relatively incremental: the challenges in extending standard attack models and solution approaches seem minor, with the extensions appearing straightforward.  This is also true for the certified robustness results, which seem rather straightforward extensions of Wong + Kolter in the first case, and Cohen et al. in the second.  Finally, experimental evaluation is not all that convincing: in Figure 1, the differences are quite small, except in Figure 1a, where the one highly effective approach is essentially a variant of Tong et al. USENIX Security 2019 "conserved features" idea [1].

Here are some further details (and see two key missing references below, esp. reference [1]).

1) Problem-space attacks: this was the terminology adopted by the Pierazzi et al. [20] (reference from paper).  Another term that means essentially the same thing "realizable attacks", which features both in security and image classification applications (e.g., [1-3] below).  The authors appear to miss this connection.

2) In Figure 1a, the best performing method is the one which masks "unmodifiable features".  This idea is similar in concept to the notion of "conserved features" from Tong et al. [1], who apply this idea also in malware classification, and also show significant performance improvement, albeit in the PDF malware domain.  It would probably be best to add PDF malware classification experiments with the same problem-space attacks as used in Tong et al., and compare to their approach as the baseline, to convincingly show that there is indeed added value to other variants, such as those based on MD, in the malware classification domains.

3) I am greatly concerned about making discrete features continuous artificially, as is done for the credit risk and spam use cases: this seems to unnecessarily handicap the feature-space approaches: after all, there are feature-space attack methods that can handle, say, binary features (e.g., JSMA, stochastic local search, etc).

4) I am also concerned that Figure 1 has no confidence intervals or other statistical information about the differences.  In most cases, the differences are quite small, and seem unlikely to be statistically significant (particularly once one adds Bonferroni correction, since we are dealing with multiple-comparisons here).  As such, the "Yes" response in 3(c) of the checklist seems extremely misleading, as Figure 1 presents the main results in the main body of the paper.

5) I don't understand what makes the attack in the spam detection use case a problem-space attack.  It seems just another variation of a feature-space attack.

6) As mentioned above, Section 4 seems to present a straightforward variation of the two state-of-the-art ways of certifying robustness (LP duality and randomized smoothing).  Moreover, these are still reliant on highly stylized assumptions embedded in \Omega, and need not certify anything against actual problem-space attacks, so the utility of these approaches in the target domains, particularly malware classification, seems limited.

[1] Tong et al. Improving robustness of ML classifiers against realizable evasion attacks using conserved features. USENIX Security, 2019.

[2] Saha and Sim.  Is Face Recognition Safe from Realizable Attacks?  IJCB 2020.

[3] Wu et al. Defending Against Physically Realizable Attacks on Image Classification. ICLR 2020.

Post-rebuttal discussion with the authors:

After the authors posted their rebuttal, I went back and reread the paper.  As a result, I increased my score, but not enough to cross into the "accept" territory.  I believe I was wrong in my original review that Section 4 has limited contribution; indeed, after reading it again I believe I had originally undervalued its significance.

My primary remaining concern about the paper, which I expressed to the authors during the somewhat involved back-and-forth discussion, is that the paper simply lacks evidence to support its main hypothesis that the proposed idea would indeed be generally effective to defend against problem-space attacks.  First, the authors clearly missed relevant prior art.  Second, two of the three attacks in the experiments are not in fact problem-space attacks, and no evidence is provided to support the claim that they are "realistic" (i.e., reasonable proxies for actual evasion attacks in those domains).  As such, I am inclined to dismiss these as provided satisfactory evidence.  The malware domain does clearly involve a problem-space attacks, but it is a very weak attack.  Realistic adversaries would make deeper changes to malware, such as removing/replacing portions.  Even aside from this, it would be crucial to demonstrate generalizability, i.e., that the approach can handle problem-space attacks of several varieties.  Consequently, I strongly recommend that the authors add experiments in the PDF malware domain, where a) there are actually reasonable state-of-the-art defense baselines, and b) lots of distinct problem-space attacks to work with, including those (like EvadeML) that add and remove content.

Post-post rebuttal:

The authors added PDF evaluation during the discussion phase, and their results are convincing to me.  I am therefore now recommending that the paper is accepted.

**Time Spent Reviewing:**

6

---

> ### Author Response · Authors · 2021-08-09
> **Response to the Reviewer 8HjK**
>
> We thank the reviewer for the detailed comments and glad that they agree with us that our approach hits an important gap.
>
> 1. We agree that nonuniform norm-ball attacks can also be realistic in the image domain, but this is usually not the case in the domains we considered in our experiments. Our focus is more on studying these underexplored domains rather than general problem space attacks. We will clarify this in the paper as well.
>
>
> 2. We thank the reviewer for the reference [1]. Our defense based on masked features is indeed similar to the approach provided by [1]. We added these results to mimic the best case defense scenario where expert’s knowledge is available. Therefore, it is not surprising that the masking approach yields the best results. However, expert knowledge may not be accessible in many applications. Our other approaches which are based on MD, Shapley and Pearson’s correlation are data-dependent and do not require such knowledge on the attack.
>
>
> 3. Our discrete features that are used in the datasets are ordinal, which makes this relaxation less concerning. But we agree that our uniform and nonuniform ATs can be improved further. Nevertheless, we applied **the same relaxation** to both uniform and nonuniform defenses in our comparisons.
>
>
> 4. We did not include confidence intervals in Fig. 1 due to visual clarity concerns, instead we referred to the Tables A.2-A.4 in the Appendix for the margins. Non-uniform approaches are significantly better than the uniform approach in all cases except the largest perturbation case in Fig 1.a. We agree with the reviewer that the margins should also be presented in the main body, hence we will change Fig. 1 accordingly.
>
>
> 5. We tested our approach for a variety of **realistic attacks** which include the problem space attacks applied to the malware binaries, as well as feature space attacks for credit-risk and spam detection. The Twitter dataset for the spam use-case is a feature representation of legitimate and bot accounts. The attack targets the samples around the decision boundary by utilizing model explainability, which makes it a realistic attack.
>
>
> 6. Our key contribution in Section 4 is to extend the use of non-uniform perturbation from empirical defenses to certification. Our certification results generalize the linear relaxations to non-uniform bounds for LP approach, and isotropy assumption to the non-isotropic covariance matrix for randomized smoothing. Our proof for the LP approach requires a non-trivial transformation to the dual problem using the Cauchy Schwarz inequality. Table 1 and 2 also show that non-uniform certification success rate significantly outperforms both existing methods. Hence, we believe our contributions add value to the literature.

---

> > ### Author Response · Authors · 2021-09-01
> > **Update for the Response to the Reviewer 8HjK**
> >
> > We would like to update our response to the reviewer's statement "Finally, experimental evaluation is not all that convincing: in Figure 1, the differences are quite small"
> >
> > We provide our new test results against various realistic attacks, such as FGSM, Carlini-Wagner, JSMA, Deep Fool and PGD in the responses to the reviewers Y8xB and tqQw. The new results show that our non-uniform approach outperforms the uniform approach in all cases with larger margins than previously presented results. Please also see our responses to the reviewers Y8xB and tqQw.

---

> > > ### Comment · Reviewer_8HjK · 2021-09-01
> > > **Re: Update for the Response to the Reviewer 8HjK**
> > >
> > > I saw these results and appreciate the effort.  However, this was actually not a concern for me, as I was primarily interested in the performance against problem-space attacks.
> > >
> > > I would also push back on the availability of expert knowledge in response to comment #1.  In all three domains the authors study, any serious application of ML would surely need expert knowledge in any case, even just to design the system that uses ML.

---

> > > > ### Author Response · Authors · 2021-09-01
> > > > **Response to the Reviewer's Comment**
> > > >
> > > > We would like to thank the reviewer for clarification on the expert knowledge. To be more specific, our masking approach represents a scenario in which the attacks are known to affect only certain malware features. We can see more examples like this in Android malware as well. In the paper, we used the term **expert knowledge** for such extra information, and presented its results as a best-case defense scenario together with our proposed approaches.

---

> > > > > ### Comment · Reviewer_8HjK · 2021-09-01
> > > > > **RE:**
> > > > >
> > > > > I appreciate the clarification.  I would suggest looking through the methodology of identifying such features automatically in the reference [1] I mention in the review.  The key message there is that it actually suffices to exclude a small subset of features from consideration (i.e., that we can be fairly confident the attacker is unlikely to be able to modify) for the same general idea to work; these are termed "conserved" features in [1].  So, what I have in mind by expert knowledge is really domain knowledge.

---

> > > > > > ### Author Response · Authors · 2021-09-02
> > > > > > **Response to the Reviewer's Comment**
> > > > > >
> > > > > > We would like to thank the reviewer for the suggestions and reference [1].
> > > > > >
> > > > > > In [1], identifying the conserved features is based on manipulating each object in each PDF file and checking the maliciousness in a sandbox. Similar to our “best case method”, [1] still needs to know the attacks that exist to build these feature sets. In our masking approach we assume that we know the immutable features, however, applying the known attacks in training time and detecting these features is trivial.
> > > > > > Though this is the best case situation and our method matches closely to the conserved features approach in [1], we do not think this approach generalizes to all defensive scenarios. It is not feasible for defenders to know all possible attack methodologies at training time or feature set generation time. This is the main reason why we presented the masking approach in the expert knowledge scenario as the best case situation which is not available all the time.
> > > > > >
> > > > > > The other non-uniform approaches we propose are generalizable to situations where the problem space attacks are not always known. It also makes sense that the other non-uniform approaches do not perform as well as the best-case approach, i.e., masking, but they still outperform uniform perturbations in all cases.
> > > > > >
> > > > > > One of the motivations of our paper is to show the need to explore several non-uniform perturbation approaches. We also think our approach is fairly generalizable as shown across our different domains in situations where you may or may not know all of the possible problem space attacks.

---

> > > > > > > ### Comment · Reviewer_8HjK · 2021-09-02
> > > > > > > **Re: Response to the Reviewer's Comment**
> > > > > > >
> > > > > > > I appreciate that the authors take the time for this back-and-forth.  However, it seems to me that the following quote is incorrect: "[1] still needs to know the attacks that exist to build these feature sets".  To the best of my understanding of [1], the approach for identifying conserved features is attack agnostic, using only the sandbox for this purpose.  Specifically, my understanding of the idea is that it essentially tries to remove particular objects (corresponding to features) from PDF, and checks whether the resulting PDF breaks malicious functionality.  This seems to only require the knowledge of the full feature space (set of relevant objects) and a sandbox to check maliciousness, but not any knowledge of actual attacks (which can make changes, including additions and removals, to collections of PDF objects in relatively arbitrary ways).  Did I miss something?
> > > > > > >
> > > > > > > However, I believe the authors are correct that the particular idea will only work for pdf malware, and would not generalize to other scenarios directly.  Nevertheless, the identification of a small subset of "immutable" features seems in principle a generalizable idea that only requires the understanding of a particular domain and feature extraction methods.
> > > > > > >
> > > > > > > In any event, my main question is this: can the authors demonstrate that their proposed idea is competitive with state of the art defense in the PDF malware domain (I wouldn't expect it to be better, as it's clearly more general), which has extensive prior literature, with many examples of problem space attacks?  This demonstration alone would be extremely compelling for the case that this paper is trying to make.

---

> > > > > > > > ### Author Response · Authors · 2021-09-02
> > > > > > > > **Response to the Reviewer's Comment**
> > > > > > > >
> > > > > > > > We would like to add a clarification regarding the reviewer’s statement “it seems to me that the following quote is incorrect:”
> > > > > > > >
> > > > > > > > Though the approach is “attack agnostic”, it requires the defenders to have access to a representative set of malicious files and run them through a sandbox system to understand the possible malicious functionality. This is at times limiting for several reasons:
> > > > > > > >
> > > > > > > > - Requires us to have a highly representative set of malicious files that represent all possible attack functionality. In general, this is a highly challenging problem and we mention in our paper this creates a very high barrier of entry into the space. We frequently see model performance decrease in many situations specifically because the input sample set is not representative of all attacks in the wild.
> > > > > > > >
> > > > > > > > - Having a perfect sandbox that can identify both maliciousness and the reasons for maliciousness is a frequently researched space. In many applications there are situations where sandboxes do not work for certain files or attackers detect sandbox environments thereby not revealing their malicious functionality.
> > > > > > > >
> > > > > > > > - Conserved features may only exist for a certain point of time. Changes in the software protocol as well as software vulnerabilities may change what is possible in an real-world attack.
> > > > > > > >
> > > > > > > > Though [1] is an ideal situation, it is not feasible to assume defenders have this knowledge at all times for all possible threat landscapes. In these situations a more generalizable approach without needing all possible malicious samples is valuable.

---

> > > > > > > > > ### Comment · Reviewer_8HjK · 2021-09-03
> > > > > > > > > **Re: response**
> > > > > > > > >
> > > > > > > > > Ah, now I understand what the authors meant, thank you for clarifying.  I think the authors are correct that the referenced approach requires malicious files as a starting point.  However, it seems to me that the proposed approach in this paper (as for any feature-space attack models) does as well.  (Just to be clear, by "attack", I mean evasion attack starting with a given malware entity; I do not mean the original malware.)  So the issue of having a representative set of original entities that are perturbed by the attacker to evade ML seems fundamental, and not something that this particular paper addresses either (at least to the best of my understanding).
> > > > > > > > >
> > > > > > > > > The authors make a point about the requirement of a perfect sandbox.  The authors are, of course, right: having a working sandbox is non-trivial.  However, as I understand, all you need from a sandbox is a binary answer whether the file is malicious or not; in any case, this is a good point.  And, of course, the authors comment about limitations of conserved features seems entirely reasonable to me.  I also completely agree that the proposed approach is clearly far more general.  This is why it view the paper as borderline: it has clear merit.
> > > > > > > > >
> > > > > > > > > What I believe it requires to decisively convince me is far stronger experimental evidence that the idea actually works for multiple examples of credible problem-space attacks in different domains (actually, I would even be satisfied if it worked in a single domain; my main criterion is generality vis-a-vis attacks).  I don't believe that any of the non-malware experiments are informative at all in this regard: perhaps subjectively, I don't think the attacks used are convincing as even reasonable proxies for problem-space attacks in those domains (there is no evidence for this provided, just intuition, and I don't share this intuition).  In the malware domain, only a single problem space attack is used, and an extremely limited one (content can only be added, but not removed/replaced).  It would be quite important to demonstrate efficacy against stronger attacks.  For this reason, I recommend the PDF malware domain, where there are indeed many examples of rather distinct problem space attacks (i.e., using different evasion strategies).  Demonstrating that the proposed approach is competitive with the state of the art would be entirely convincing to me (I emphasize that being competitive is sufficient; being much worse would perhaps be less convincing).

---

> > > > > > > > > > ### Author Response · Authors · 2021-09-06
> > > > > > > > > > **Response to the Reviewer: PDF Malware Results**
> > > > > > > > > >
> > > > > > > > > > We would like to thank the reviewer for the recommendations.
> > > > > > > > > >
> > > > > > > > > > We tested NU-$\delta$-MDtarget and Uniform-$\delta$ against EvadeML for PDF malware as the reviewer suggested. We used the pdf dataset with 110841 samples and 135 features extracted by PDFrate-R which is provided in https://github.com/csmutz/pdfrate. We obtained the best performances at $||\delta||_2$=1 case for both methods. The table below depicts the accuracy, AUC score and defense success rate (DSR) against EvadeML for standard training (no defense), Uniform-$\delta$ and NU-$\delta$-MDtarget.
> > > > > > > > > > These results show that, as before, our non-uniform approach is effective against problem space attacks and outperforms the uniform approach. Although it does not reach the performance reported in Fig.4 in [1], please note that our approach does not assume any attack knowledge and hence it is more generalizable.
> > > > > > > > > >
> > > > > > > > > >
> > > > > > > > > >
> > > > > > > > > >
> > > > > > > > > >
> > > > > > > > > > |  Model                                   | Accuracy | AUC score | EvadeML DSR |
> > > > > > > > > > |------------------------------------------|----------|-----------|------------------------------|
> > > > > > > > > > | Standard training                        | 99.52%   | 0.99912   | 11.1%                        |
> > > > > > > > > > | Uniform-$\delta$ at $\|\|\delta\|\|_2$=1 | 97.64%   | 0.99826   | 87.4%                        |
> > > > > > > > > > | NU-$\delta$-MDt$ at \|\|\delta\|\|_2$=1  | 97.83%   | 0.99835   | 92.9%                        |

---

> > > > > > > > > > > ### Comment · Reviewer_8HjK · 2021-09-06
> > > > > > > > > > > **Re: PDF Malware Results**
> > > > > > > > > > >
> > > > > > > > > > > Excellent -- this is enough to convince me.  If the paper is ultimately accepted, I recommend adding evaluation with 1 more attack (e.g., the Mimicry attack).  I would also recommend a somewhat deeper dive into prior work on realizable attacks in related work, to include physical and physically realizable attacks/defenses in computer vision (including [3] in the review I provide, and this paper on certified defense against patch attacks: http://people.eecs.berkeley.edu/~daw/papers/minority-simla20.pdf).

---

### Official Review · Reviewer_Y8xB · 2021-07-11

**Rating:** 6
**Confidence:** 4

**Summary:**

The authors present a novel strategy of adversarial attacks with non-uniform perturbation, and provide certified robustness for the new norm-bounded perturbation strategy. Both theoretical and empirical studies are presented to support the consideration of feature importance in attacks and adversarial training.

**Limitations And Societal Impact:**

The authors have made significant improvement over their ICML manuscript. However, I have several concerns about the current submission:

I don't quite agree with the claim that small norm-bounded perturbations are necessary for maintaining imperceptibility. Sharma & Chen in "Attacking the Madry Defense Model with $L_1$-based Adversarial Examples" clearly demonstrate that high average norm-bounded distortions may have minimal visual distortion. In other words, adversaries may have a much greater de facto attack budget than certified models have assumed. The technique presented in this paper is also norm-bounded, therefore is susceptible to this misconception.

Human perceptual systems do not share the same metamers as machine learning models, therefore, attacks in-out a given norm-bound may be equally capable of deceiving a human as long as they have similar aggregate statistics. See:
[1.] "Exploiting Human Perception for Adversarial Attacks", by Pengrui Quan and Mani B. Srivastava
[2.] "Metamers of neural networks reveal divergence from human perceptual systems", by Jenelle Feather, Alex Durango, Ray Gonzalez, Josh McDermott in NeurIPS 2019.
This applies to tabular data as well. It is unclear what a universally realistic budget should be for a given problem. This calls into the question the robustness of certified models. Therefore, in the test phase of the empirical study, adversaries should not be constrained with a given budget. In addition, models certified for one type of attack should be tested on different types of attacks.

The proposed technique takes advantage of given domain knowledge. For fair comparison, the Madry model with uniform perturbation should be given the same advantage by enabling feature weighting, for example, assigning an a-priori importance value as in "Learning to classify with missing and corrupted features" by Dekel and Shamir, ICML2008. A simple feature scaling on $\delta$ in each iteration of attack should also be sufficient.

It's not clear how many iterations have been performed during the optimization-based attack. The defense success rates of the standard training presented in the Appendix are pretty high for the Malware and Credit Risk datasets.


**Main Review:**


Cost, utility, and feature importance are reasonable concerns for adversaries in practice. This paper tackles the challenge of optimizing attacks with non-uniform perturbation, in which features are not treated equally. Non-uniform perturbations appear to be more effective in the empirical studies on three real datasets. Robust certification techniques based on non-uniform perturbation are also provided, which completes an attack-defense problem cycle.

**Time Spent Reviewing:**

approximated 5-6 hours

---

> ### Author Response · Authors · 2021-08-09
> **Response to the Reviewer Y8xB**
>
> - Imperceptibility
>
> We agree that a norm-bounded attack with a large perturbation budget can still be imperceptible in the image domain. In the paper, we emphasize that the same amount of perturbation budget during adversarial training (AT) provides higher robustness when a non-uniform norm-ball constraint is used, and the resulting perturbations would still be imperceptible. Therefore, we will eliminate any misunderstanding caused by the wording “small perturbation” in the paper. Besides, uniform perturbation constraint which works well in the image domain provides limited robustness for the underexplored domains such as malware, spam detection and fraud detection.
>
> - Models certified with one type of attack should be tested on different attacks
>
> We would like to thank the reviewer for the references. We agree that there is a lack of universally realistic budget for AT. As the reviewer suggests, we tested with unbounded budget real-world malware attacks. The only constraints of these attacks were real-world limitations such as file size limit, and feasibility. We also tested with norm-bounded feature space attacks for spam and credit risk detection use-cases to explore a wider range of realistic attacks. We **indeed** tested models certified with one type of attack on different attacks as suggested. Table 1 shows the defense success rates of **Uniform-$\delta$** and **NU-$\delta$-MDt** against the spam detection attack of Section 3.3, whereas these models are certified with the approaches listed under **Cert. Method**. In Table 2, similar certification results are shown, and the defense success rates can be seen in Table A.4.
>
> - Domain knowledge
>
> Our proposed approach mostly uses data-dependent knowledge, i.e., SHAP values, Pearson correlations, Mahalanobis distances. We used domain knowledge only when we mask immutable features in malware use-case. Enabling feature weighting with apriori importance value for Madry model would be transforming Madry’s to a non-uniform approach which essentially what our work proposes by NU-$\delta$-Pearson and NU-$\delta$-SHAP. We performed additional experiments to support fairness, and tested Uniform-$\delta$ and NU-$\delta$-MDtarget against uniform PGD attacks. Please see the table in the response to the Review tqQw.
>
> - High defense success rate
>
> High defense success rates show the strength of the attacks. However, in malware use-case, it is due to averaging the performance over several strong and weak attacks. Weak attack performances dominate the average defense success rate in Table A.2.. Please see Table A.1 for the evasion success of each individual malware attack.

---

> > ### Comment · Reviewer_Y8xB · 2021-08-15
> > **Regarding the authors' response**
> >
> > Thank you for the clarifications.
> >
> > Regarding the perturbation budget, in your response to the Reviewer tqQw, when delta is large (0.5), it is hardly convincing that "the non-uniform approach is more robust than the uniform approach even against uniform PGD attacks".  Can you provide statistical significance as you did in the Appendix for all results in the table? The results shown in the Appendix do not suggest statistical consistency favoring one  specific non-uniform approach. A general claim that a non-uniform approach is better is a bit of a stretch.
> >
> > Regarding testing certified models on "different types of attacks", what I meant is the underlying attack technique such as Carlini & Wagner attack, Fast Gradient Method, DeepFool, Jacobian Saliency Map Attack, etc. instead of just PGD attacks. You don't need to test against all, but at least one or two should be considered. In the wild, a certified model would have no idea what the underlying attack method it would encounter.
> >
> > Regarding domain knowledge, is it terribly difficult to "transform" Madry's approach to include expert knowledge such as marking features immutable or performing feature importance analysis before attack? Is the sole merit of the proposed non-uniform approach "knowing" something while assuming other techniques could not possibly know?
> >
> > Regarding high defense success rate, more severe attacks should be investigated by allowing higher perturbation budgets.

---

> > > ### Author Response · Authors · 2021-08-17
> > > **Response to the Reviewer's Comment**
> > >
> > > - Statistical significance for the table in the response to the Reviewer tqQw
> > >
> > > We are sharing an updated version of the table in the response to the Reviewer tqQw below. We also tested the defenses with larger perturbations $||\delta||_2=1$  and $||\delta||_2=1.5$, and observed that the differences between the defense success rates of the uniform and non-uniform approaches become larger for larger perturbations as the attack becomes stronger.
> > > When we compare the best performances of both defenses, the non-uniform approach outperforms the uniform approach significantly at $||\delta||_2=1.5$, i.e., %89.78 for Uniform-$\delta$ and %93.48 for NU-$\delta$-MDt.
> > >
> > > |  Defenses            | $\delta$ | Uniform Attack $\epsilon$=0.1 | Uniform Attack $\epsilon$=0.3 | Uniform Attack $\epsilon$=0.5 | Uniform Attack $\epsilon$=0.7 |
> > > |----------------------|----------|-------------------------------|-------------------------------|-------------------------------|-------------------------------|
> > > | Uniform-$\delta$     | 0.1      |     $90.6 \pm 0.18$           |     $83.9 \pm 0.34$           |     $16.6 \pm 0.47$           |     $14.8 \pm 0.65$           |
> > > | NU-$\delta$-MDtarget | 0.1      |     $92.8 \pm 0.21$           |     $88.2 \pm 0.41$           |     $34.95 \pm 0.5$           |     $21.4 \pm 0.22$           |
> > > | MDtarget-Combo       | 0.1      |     $91.6 \pm 0.24$           |     $86.3 \pm 0.28$           |     $24.4 \pm 0.38$           |     $19.2 \pm 0.32$           |
> > > | Uniform-$\delta$     | 0.3      |     $92.6 \pm 0.14$           |     $89.05 \pm 0.24$          |     $30.5 \pm 0.42$           |     $20.85 \pm 0.58$          |
> > > | NU-$\delta$-MDtarget | 0.3      |     $93.2 \pm 0.10$           |     $90.95 \pm 0.22$          |     $61.75 \pm 0.51$          |     $46.3 \pm 0.41$           |
> > > | MDtarget-Combo       | 0.3      |     $93 \pm 0.11$             |     $89.24 \pm 0.19$          |     $47.88 \pm 0.72$          |     $31.5 \pm 0.39$           |
> > > | Uniform-$\delta$     | 0.5      |     $92.9 \pm 0.08$           |     $91.4 \pm 0.05$           |     $88 \pm 0.22$             |     $86.5 \pm 0.25$           |
> > > | NU-$\delta$-MDtarget | 0.5      |     $93.3 \pm 0.05$           |     $91.45 \pm 0.06$          |     $89.45 \pm 0.17$          |     $87.7 \pm 0.12$           |
> > > | MDtarget-Combo       | 0.5      |     $93.1 \pm 0.10$           |     $91.4 \pm 0.11$           |     $89.30 \pm 0.14$          |     $87.2 \pm 0.18$           |
> > > | Uniform-$\delta$     | 1        |     $93.2 \pm 0.14$           |     $91.9 \pm 0.25$           |     $90.18 \pm 0.20$          |     $88.38 \pm 0.22$          |
> > > | NU-$\delta$-MDtarget | 1        |     $94.5 \pm 0.11$           |     $94.17 \pm 0.42$          |     $93.33 \pm 0.15$          |     $93.14 \pm 0.31$          |
> > > | MDtarget-Combo       | 1        |     $94.39 \pm 0.10$          |     $92.42 \pm 0.28$          |     $91.44 \pm 0.19$          |     $91.33 \pm 0.26$          |
> > > | Uniform-$\delta$     | 1.5      |     $93.22 \pm 0.16$          |     $92.01 \pm 0.27$          |     $90.61 \pm 0.24$           |     $89.98 \pm 0.15$          |
> > > | NU-$\delta$-MDtarget | 1.5      |     $94.45 \pm 0.12$          |     $94.38 \pm 0.31$          |     $93.76 \pm 0.16$           |     $93.48 \pm 0.11$          |
> > > | MDtarget-Combo       | 1.5      |     $94.27 \pm 0.11$          |     $93.1 \pm 0.20$           |     $92.3\pm 0.27$             |     $91.8 \pm 0.17$           |
> > >
> > > - Results do not suggest a single non-uniform approach
> > >
> > > When the best performances are considered, malware and spam detection use-cases favor Mahalanobis distance approach, and credit risk use-case favors feature importance-based methods. This is due to the fact that the LowProFool attack used in credit risk detection targets the most important features by using an importance regularization, hence our feature importance-based defense outperforms the rest against feature importance-based attacks.
> > >
> > > Our motivation is **enabling** non-uniform perturbations by $\Omega$ selection, and proposing **effective** $\Omega$’s by utilizing domain or attack knowledge. Though all non-uniform approaches are effective, when expert knowledge is not available the results favor Mahalanobis distance approach in most cases.
> > >
> > > - Testing against different attacks such as Carlini & Wagner attack, Fast Gradient Method, DeepFool, Jacobian Saliency Map Attack, etc. instead of just PGD attacks
> > >
> > > To clarify, we evaluated our approach with real problem space attacks in malware use-case, LowProFool in credit risk use-case, and an interpretability-based attack that prioritizes attacking salient samples in spam detection use-case. These are not standard PGD attacks. To address the reviewer’s comment, we provide additional defense success rate evaluations for the certified defenses in Section 4 with the suggested attacks by the reviewer using the Adversarial Robustness Toolbox. We used the default parameters for the AE generators of Carlini Wagner, JSMA, and DeepFool Methods. We also crafted AEs using FGSM for $\epsilon=0.5$ and $\epsilon=1$. In the table below, we observe that NU-$\delta$-MDt outperforms the uniform approach in all cases.
> > >
> > > | Defenses         | $\|\|\delta\|\|_2$ |Benign Clean Acc. | FGSM $\epsilon$=0.5 | FGSM $\epsilon$=1 | Carlini Wagner   | JSMA             | Deep Fool        |
> > > |------------------|--------------------|-----------------------------------|---------------------|-------------------|------------------|------------------|------------------|
> > > | Uniform-$\delta$ | 0.1                | 91.61                             | $24.1 \pm 0.25$     | $20.17 \pm 0.31$  | $38.93 \pm 0.54$ | $41.51 \pm 0.24$ | $39.3 \pm 0.27$  |
> > > | NU-$\delta$-MDt  | 0.1                | 91.93                             | $28.7 \pm 0.17$     | $27.1 \pm 0.21$   | $49.62 \pm 0.33$ | $49.59 \pm 0.18$ | $51.73 \pm 0.31$ |
> > > | Uniform-$\delta$ | 0.3                | 90.40                             | $25.22 \pm 0.12$    | $21.61 \pm 0.43$  | $45.83 \pm 0.41$ | $46.38 \pm 0.27$ | $47.39 \pm 0.25$ |
> > > | NU-$\delta$-MDt  | 0.3                | 91.31                             | $32.15 \pm 0.34$    | $30.88 \pm 0.36$  | $53.45 \pm 0.22$ | $52.68 \pm 0.13$ | $54.61 \pm 0.19$ |
> > > | Uniform-$\delta$ | 0.5                | 87.12                             | $27.05 \pm 0.50$    | $23.08 \pm 0.34$  | $50.05 \pm 0.32$ | $49.5 \pm 0.25$  | $49.75 \pm 0.16$ |
> > > | NU-$\delta$-MDt  | 0.5                | 87.78                             | $43.85 \pm 0.62$    | $36.12 \pm 0.20$  | $55.24 \pm 0.38$ | $53.71 \pm 0.41$ | $64.28 \pm 0.55$ |
> > > | Uniform-$\delta$ | 1                  | 86.46                             | $40.20 \pm 0.57$    | $32.98 \pm 0.31$  | $52.77 \pm 0.24$ | $52.94 \pm 0.26$ | $53.22 \pm 0.10$ |
> > > | NU-$\delta$-MDt  | 1                  | 87.64                             | $81.34 \pm 0.83$    | $79.85 \pm 0.75$  | $61.89 \pm 0.37$ | $72.93 \pm 0.77$ | $88.75 \pm 0.78$ |
> > > | Uniform-$\delta$ | 1.5                | 85.98                             | $74.40 \pm 0.47$    | $62.36 \pm 0.75$  | $59.71 \pm 0.28$ | $68.95 \pm 0.85$ | $64.5 \pm 0.86$  |
> > > | NU-$\delta$-MDt  | 1.5                | 87.03                             | $94.45 \pm 0.18$    | $91.03 \pm 0.10$  | $72.98 \pm 0.48$ | $86.43 \pm 0.32$ | $98.36 \pm 0.25$ |
> > >
> > > - Domain knowledge
> > >
> > > Marking immutable features and not using the perturbation budget for those features in the Madry model would be similar to suggesting non-uniform perturbations as we proposed. Since immutable and important feature concepts are entirely different, a feature importance analysis before the attack does not serve the same purpose. For instance, in the malware domain, a feature which is immutable due to the software feasibility might still be important for malware analysis.
> > >
> > > - High defense success rate
> > >
> > > Malware use-case attacks that we crafted are unbounded since they are in the problem space. For further investigation of stronger attacks, we tested our defenses against the attacks suggested by the reviewer, such as FGSM, Carlini Wagner, JSMA and Deep Fool. We provided the results in the table above.

---

> > > > ### Comment · Reviewer_Y8xB · 2021-08-19
> > > > **Thank you for the clarification**
> > > >
> > > > When delta increases in the defense, what's the accuracy loss on the unattacked positive samples?
> > > >
> > > > Can you also provide the results by setting delta = 0.1 and 0.3 when compared to FGSM, CW and DeepFool?

---

> > > > > ### Author Response · Authors · 2021-08-20
> > > > > **Response to the Reviewer's Comment**
> > > > >
> > > > > Thanks to the reviewer’s comment, we updated the results for FGSM, CW, JSMA and Deep Fool attacks (the second table in our previous response) by including **Benign Clean Acc.** which is the clean accuracy of the benign samples (unattacked samples), and defense success rates for $||\delta||_2=0.1$ and $||\delta||_2=0.3$ cases.
> > > > >
> > > > > Clean accuracy rate decreases as $||\delta||_2$ increases both for uniform and non-uniform approaches but the degradation in non-uniform is  less. Defense success rate, on the other hand, improves for both approaches but our non-uniform approach significantly outperforms the uniform approach.

---

> > > > > > ### Comment · Reviewer_Y8xB · 2021-08-26
> > > > > > **Thank you for your reply**
> > > > > >
> > > > > > My questions have been addressed. Thank you.

---

### Official Review · Reviewer_LM6b · 2021-07-15

**Rating:** 6
**Confidence:** 3

**Summary:**

The authors study how to improve adversarial training by leveraging non-uniform perturbations of the input space.
The latter are formalised by considering non-isotropic mutations of the input sample, and this is useful for addressing domains like the malware one.
Experimental results show that adversarial training with non-uniform perturbation is more robust than regular adversarial training, by offering different use cases (malware, credit risk, spam detection).
The authors conclude the work by certifying the robustness of their approach.

**Limitations And Societal Impact:**

The authors did not discussed the limitations of their work, but there are some that are highlighted inside the review. The authors might start from them.

**Main Review:**

Pros:
* the intuition of using non-isotropic perturbation is clear and very useful, especially in domains where data are more complex (e.g. malware)
* the authors tested many different domains to prove the efficacy of their intuitions
* the adversarial training results are certifiable

Cons:
* the contributions of this work are incremental, and novelty seems very small compared to the original work [1, 2]. Also, the authors would need to state that there are many other domains that might benefits from non-uniform perturbations (like audio, text, android malware [3-6]).
* Concerning the attacks used for the adversarial training pipeline for the malware domain, it is unclear which manipulation has been used. Also, there is lack of the state-of-the-art manipulations. Authors may need to consider the latest work that address the generation of adversarial malware [7].
* While the results looks very good, there is the concern regarding the implementation of the submitted code.
First, the implementation of the PGD attack is custom. Why the authors did not use any known library, like Foolbox, Adversarial Robustness Toolkit (ART), Cleverhans, or SecML?
Furthermore, by looking at the code, it seems that the adversarial examples are not crafted on the trained model, but rather they are created while training the classifier itself. This seems a bit odd, and maybe the authors would need to comment this choice.
* Since the authors proposed an hardening of adversarial training, with certifiable bounds, it would be insightful to test your defenses with adaptive attacks that consider also the presence of such defence.
An example for an attack that you might want to comment on is [8], who crafted spoofed certificates for randomized smoothing.

Bibliography
* [1] Xu, Kaidi, et al. "Structured adversarial attack: Towards general implementation and better interpretability." ICLR 2019.
* [2] Chen Liu, et al. "On certifying non-uniform bounds against adversarial attacks." ICML 2019
* [3] Grosse, Kathrin, et al. "Adversarial examples for malware detection." European symposium on research in computer security. Springer, 2017.
* [4] Zhang, Guoming, et al. "Dolphinattack: Inaudible voice commands." Proceedings of the 2017 ACM SIGSAC Conference on Computer and Communications Security. 2017.
* [6] Roy, Nirupam, Haitham Hassanieh, and Romit Roy Choudhury. "Backdoor: Making microphones hear inaudible sounds." Proceedings of the 15th Annual International Conference on Mobile Systems, Applications, and Services. 2017.
* [7] Demetrio, Luca, et al. "Adversarial EXEmples: A survey and experimental evaluation of practical attacks on machine learning for windows malware detection." arXiv preprint arXiv:2008.07125 (2020).
* [8] Ghiasi, Amin, Ali Shafahi, and Tom Goldstein. "Breaking certified defenses: Semantic adversarial examples with spoofed robustness certificates." ICLR, 2020.


After respose:
I'll rise my score to 6 after discussion and after having read the answers.

**Time Spent Reviewing:**

4 hours

---

> ### Author Response · Authors · 2021-08-09
> **Response to the Reviewer LM6b**
>
> We would like to thank the reviewer for the comments and the suggested references. We will mention the domains like speech, text, android malware [3-6] and other references in the paper.
>
> - Comparison to [1,2] and other domains
>
> [1] proposes a strongly image domain dependent solution, which is quite different from our work. CNN-like masking that slides through the image assumes locality, which is true for images but may not be valid in the domains we considered. Our method uses feature dependencies and importance inherently due to the $\Omega$ matrix, instead of grouping the features discretely as in [1]. Therefore, we did not consider [1] as a related work.
>
> Although both our work and [2] propose non-uniform certification bounds, they are rather complementary to each other. While we certify the regions that a realistic attacker can target, [2] solves another optimization for learning $\epsilon$ budgets regardless of the importance of the regions for such an attacker. Relaxing the realistic attacker assumption brings additional computational burden to [2]. In fact, a similar approach to our $\Omega$ transformation is mentioned in [2] as a potential extension under future works in Section 5.2.
>
> To demonstrate our claims further, we certified our robust models using [2] for the setting in Fig 1.c for $||\delta||_2=0.3$. We calculated the geometric mean over learned $\epsilon$’s as:
>
> **Uniform-$\delta$** = 0.0601, **NU-$\delta$-MDt** = 0.0617, **NU-$\delta$-Pearson** = 0.0608. As a result, [2] also certifies that our nonuniform approach outperforms the uniform robustness.
>
> Overall, we propose a complete approach containing both empirical defenses and certifications, hence, we believe our approach adds significant value to the literature.
>
> - State-of-the-art problem space malware attacks
>
> Malware attacks that we used for testing are described in Appendix A.6. These are actually in the same category as the padding and benign section injection attacks in [7]. For further comparison, we also applied the SHIFT attack in [7] on 500 samples from the EMBER malicious test set. Defense success rates of the standard training, Uniform-$\delta$ and NU-$\delta$-MDt for $||\delta||_2=1$ are **74.5%**, **82.5%** and **85.6%**, respectively. Once again we show that our nonuniform approach outperforms the uniform case.
>
> - Submitted code and adversarial examples generated during training
>
> We do not have a particular reason for using custom pgd except its simple implementation. Generating the adversarial examples during training is due to the definition of adversarial training itself. Please see [A]
>
> [A] Madry, A., Makelov, A., Schmidt, L., Tsipras, D., & Vladu, A. (2018). Towards Deep Learning Models Resistant to Adversarial Attacks. ArXiv, abs/1706.06083.
>
> - Testing our approach on [8]
>
> Our contributions are in two different categories. In Section 2, we propose an empirical defense, whereas in Section 4, we propose certification methods to certify a robust model instead of proposing another defense by hardening the adversarial training.
> We thank the reviewer for the reference [8] but it is not applicable to our use-cases as it uses image specific loss functions such as smoothness and similarity between color channels which do not apply to the datasets we used.

---

> > ### Comment · Reviewer_LM6b · 2021-08-25
> > **Thank you for the answer**
> >
> > I have raised my score after having read the response.
> > Thank you for the clarifications.

---

> > > ### Author Response · Authors · 2021-08-25
> > > **Response to the Reviewer's Comment**
> > >
> > > We would like to thank the reviewer again for their careful review, suggestions, and references.

---

### Official Review · Reviewer_9LNj · 2021-07-15

**Rating:** 7
**Confidence:** 5

**Summary:**

The paper proposes a notion of robustness w.r.t. non-uniform adv. perturbation scaled according to Mahalanobis Distance (MD). By harnessing non-uniform perturbation AE and using them for adv. training, we can obtain a more robust model with both better certified and empirical robustness.

**Ethical Concerns:**

No concern.

**Limitations And Societal Impact:**

Yes, they are adequately addressed.

**Main Review:**

I like the intuition behind the paper: not all directions equally matter for classification, and we can focus on the more important ones for robustness.

Writing-wise, the paper is on the dense end with a lot of details. This might be a result of answering previous review questions. It can be improved by adding a paragraph header for similar components such as "choice of MD/transformation matrix", etc.

The question I want to understand the most, and the most critical to my final recommendation of the paper, is the choice of attack budget \delta in the experiment. When you compute the attack success rate against AT models, are the feasible set of AE the same for both uniformly-perturbed and non-uniformly-perturbed AT models?  I'm asking because AT should **not** affect the attacker's set of choice of AE. So I want to make sure that the attack feasibility set is the same.

An optional comparison result I'd like to see is comparing AT with a preprocessing based defense mechanism: the learner learns the same transformation matrix used in non-uniform perturbation, scales all input using the transformation, and eventually trains a new model over the scaled inputs. It is natural to compare AT with a direct dimension reduction fashion technique that potentially reduces adv. effect over not-so-important direction.

Last, for the malware detection task, the attack is said to be done on the problem-space. However, EMBER only contains feature vectors. Have you actually created any legit PE files in the attack? If not, please clarify that in the paper. I don't mind assuming certain virtual transformations that most likely can compile and have easy-to-see effect on the feature space. In fact, I won't reject the paper even if the attack is only on feature space for a learning conference. But please make it clear if any actual PE or executables are generated. A proper problem-space attack needs those.

Overall, I believe this paper is of good quality. I'm giving a score of 5 for now, but I'm willing to raise the score once my questions are addressed.

============== After author response ========================

My questions are addressed. Thank you. I changed my score from 5 to 7.

**Time Spent Reviewing:**

3

---

> ### Author Response · Authors · 2021-08-09
> **Response to the Reviewer 9LNj**
>
> - Feasible set of AE
>
> If the reviewer asks whether both uniform and nonuniform robust models are tested against the same set of AEs for fair comparison, that is true.
> On the other hand, if the question is whether the AT is performed for the same $||\delta||_2$ budget for both uniform and nonuniform cases, that is also true and this is what is meant by  $||\delta||_2$ in Tables A.1-A.4. Hence, the feasible sets of AEs are the same for both models.
>
> - Optional comparison to pre-processing based
>
> We thank the reviewer for the suggestion. For the experimental setting of the Section 3.3, we performed pre-processing based experiments as the reviewer suggested. Utilizing the standard PCA whitening of scikit-learn library, we reduced Twitter dataset dimension from 31 features to 20 and 10.
> The (defense success rate)-(clean accuracy) pairs of the models trained with feature size 20 and 10 are **28.5%-93.5%** and **72%-90.6%**, respectively. Due to the decrease in the attacker’s degree of freedom in manipulating a sample, such pre-processing provides a certain level of robustness. However, adversarial training is still a better choice regarding the trade-off between clean and adversarial accuracy (See Table A.4).
>
> - Problem-space attack on EMBER
>
> The malware attacks we used for testing our defenses are **indeed** problem space attacks. Although the EMBER dataset contains vectorized features of real goodware and malware binaries, these PE files can be downloaded from VirusTotal by premium account holders using the SHA256 identifier. We obtained the real malware binaries, and applied byte padding in the overlay and benign section injecting attacks. We provide a featurized version of the adversarial examples in our supplementary material.

---

> > ### Comment · Reviewer_9LNj · 2021-08-09
> > **Thanks for your clarification**
> >
> > My questions are addressed. I raised my score to 7.

---

> > > ### Author Response · Authors · 2021-08-12
> > > **Response to the Reviewer's Comment**
> > >
> > > We would like to thank the reviewer again for their careful review and suggestions.

---

> > ### Comment · Reviewer_9LNj · 2021-08-17
> > **One Additional Question**
> >
> > For malware detection, the \Omega is a simple mask covering the immutable features. Can you give a pointer to how these immutable features are determined, and why the Shapley values selection technique is not used in this setting for the mutable features? I'm asking because binary padding without preprocessing can almost flip all n-gram features from 0 to 1.

---

> > > ### Author Response · Authors · 2021-08-17
> > > **Response to the Reviewer's Additional Question**
> > >
> > > Among our malware defenses, only NU-$\delta$-Mask contains a masking $\Omega$. The rest of the $\Omega$’s are determined by Mahalanobis distance, Shapley values, and Pearson’s correlation coefficients, and are not masked.
> > >
> > >
> > > Since we do not use n-gram features, the reviewer’s assumption about binary padding does not hold.
> > >
> > > Our malware features are in two groups; parsed features, such as **General File Information**, **HeaderInformation**, **Imported Functions**, **Exported Functions**, and **Section Information**, and format-agnostic features, such as **Byte Histogram**, **Byte Entropy Histogram**, **String Information** as we described in Appendix A.2. We observed that binary padding attacks affect only the feature groups **Byte Histogram**, **Byte Entropy Histogram**, and **Section Information**. Hence for NU-$\delta$-Mask, we assume we know the attack is a binary padding attack and therefore masking is applied only to the immutable features. Please note, we do not make this assumption for the rest of our non-uniform defenses which utilize Mahalanobis distance, Shapley values, and Pearson’s correlation coefficients.

---

### Decision · Program_Chairs · 2021-09-28

**Decision:**

Accept (Poster)

**Comment:**

The paper considers an understudied and yet important area of adversarial ML.  The paper is very well written.  The need to consider attacks adjusted to specific features is intuitive and makes sense. The paper considers real world constraints and evaluates on multiple datasets of distinct domains such as malware, bank fraud and spam. The results showing how the attacks end up looking more like benign samples is compelling, and the extension of the results to certifiable bounds in two different ways is good. After extensive discussions, the added new experiments convinced the reviewers to accept the paper.

**Consistency Experiment:**

NeurIPS has a long history of experimentation. In 2014, NeurIPS ran an experiment in which 10% of submissions were reviewed by two independent committees to quantify the randomness in the review process. This year, we repeated a variant of this experiment to see how the quality of the review process has changed over time.  This paper was part of the experiment and was therefore assigned to two committees (consisting of reviewers, an Area Chair, and a Senior Area Chair) that reached independent decisions.  If both committees made the same recommendation, this recommendation was followed. If a single committee recommended acceptance, the paper was accepted (with the exception of a few cases in which the other committee identified what we considered a fatal flaw, e.g., an error in a key result).

This copy’s committee reached the following decision: **Accept (Poster)**

The other committee assigned to the paper recommended **Reject**.  You can find the other set of reviews, along with any follow up discussion with the authors here:
https://openreview.net/forum?id=oi08QWKs84